

**Influence of boundary layer structure on air quality in Beijing: Long-term**
**analysis based on self-organizing maps**
Zhiheng Liao[a], Jiaren Sun[a, b]*, Jialin Yao[c], Li Liu[a], Haowen Li[a], Jian Liu[a], Jielan Xie[a], Dui Wu[d], Shaojia Fan[a]*
[a] School of Atmospheric Sciences, Sun Yat-sen University, Guangzhou, Guangdong, China;
[b] South China Institute of Environmental Sciences, Ministry of Environmental Protection of the People's Republic
of China, Guangzhou, Guangdong, China;
[c] Weather Modification Office of Shanxi Province, Taiyuan, Shanxi, China;
[d] Institute of Mass Spectrometer and Atmospheric Environment, Jinan University, Guangzhou, Guangdong,
China.
* Address correspondence to S. Fan or J. Sun, School of Atmospheric Sciences, Sun Yat-sen University,
Guangzhou, Guangdong, China. Telephone: +86 020 8411 5522.
E-mail: eesfsj@mail.sysu.edu.cn (S. Fan); sunjiaren@scies.org (J. Sun).
**Abstract**
Self-organizing maps (SOMs; a feather-extracting technique based on an unsupervised machine learning
algorithm) are used to classify the atmospheric boundary layer (ABL) types over Beijing by detecting topological
relationships among the 4-yr (2013–2016) radiosonde profiles. The resulting ABL types are then examined in
relation to air quality, including surface pollutant concentrations and columnar aerosol properties, to understand
the regulating effects of different ABL structures on Beijing's air quality. The SOM provides nine ABL types (i.e.,
SOM nodes), and each type is characterized by distinct dynamic and thermodynamic conditions. On average, $SO_2$,
$NO_2$, CO, $PM_{10}$ and $PM_{2.5}$ increase 120–220 % from a near neutral (i.e., node 1) to strong stable condition (i.e.,
node 9). The ABL controls on diurnal cycles of pollutants are as follows: (1) elevated inversion enhances the
afternoon baseline; and (2) surface inversion improves the evening increment. Comparing the $CO/SO_2$ ratios for
the different ABL types demonstrates that the local contribution increases with enhanced static stability near the
ground, and it is the stable ABL stratification rather than weak surface wind that confines the regional contribution.
Due to regional transport, node 3 (dominated by elevated inversion with high relative humidity) corresponds to
the most severe columnar aerosol pollution, characterized by the highest optical depth (1.22) and volume
concentration (0.30 $\mu m^3/\mu m^2$). The larger aerosol radiative forcing (ARF) within the atmosphere (> 60 $W/m^2$) in
nodes 3, 6 and 9 is likely to strengthen the atmospheric stability and thus induce a positive feedback loop for



causing high surface pollution. Analysis of the typical pollution period suggests that the ABL types are the
primary drivers of day-to-day variations in Beijing's air quality. Assuming a fixed relationship between ABL type
and $PM_{2.5}$ loading for different years, the relative (absolute) contribution of the ABL anomaly to elevated $PM_{2.5}$
levels are estimated to be 65.8 % (46.2 µg/m$^3$) during January 2013, 46.7 % (20.2 µg/m$^3$) during December 2015,
and 94.6 % (35.3 µg/m$^3$) during December 2016.

**1 Introduction**

The atmospheric boundary layer (ABL) is the section of atmosphere that responds directly to the flows of mass,
energy and momentum from the earth's surface, characteristically at timescales of an hour or less (Stull, 1988).
Most air pollutants are emitted or chemically produced within this layer and its evolution plays an important role
in determining the dispersive and chemical properties of pollutants (Chen et al., 2012; Fan et al., 2008; Whiteman
et al., 2014; Platis et al., 2016; Wolf et al., 2014; Wu et al., 2013). Therefore, characterizing typical ABL
conditions associated with high pollution levels helps to better understand the role of ABL in governing the
transport and distribution of pollutants in the atmosphere.

Beijing, the capital of China, is suffering serious air pollution problems. This city is geographically located at
the northwestern border of the Great North China Plain and has three directions that are adjacent to mountains.
The closest coast from the city of Beijing is the Bohai Sea, which is 160 km southeast of the city. Terrain-related
circulations can therefore be well developed over Beijing and its surroundings under favorable weather conditions,
leading to a complex ABL thermodynamic structure, which is thought to substantially affect Beijing's air quality
(Hu et al., 2014; Miao et al., 2017; Ye et al., 2016; Gao et al., 2016; Xu et al., 2016). Several studies used
tower-based observations to investigate the interactions between boundary layer dynamics and pollution formation
(Sun et al., 2013; Sun et al., 2015; Guinot et al., 2006). However, the results are not ideal because they have a low
observational height (325 m). Numerous intensive ABL measures were conducted using other approaches, such as
mooring boats, airplane, and ground remote sensing (Tang et al., 2015; Zhu et al., 2016; Zhang et al., 2009; Hua et
al., 2016). However, since these approaches are complex, expensive and labor intensive, they are often restricted
to the duration of specific research campaigns. Overall, the existing knowledge of linkages between ABL structure
and air quality in Beijing is drawn largely from either low observational height or short observational duration.
Due to the lack of long-term effective observations, the influence of ABL on Beijing's air quality remains
relatively unclear. For example, many case studies (Jia et al., 2008; Zheng et al., 2015; Hua et al., 2016; Li et al.,
2016) claimed that rapid growth of $PM_{2.5}$ in Beijing is mainly attributable to the regional transport of the polluted



air mass. This view is occasionally questionable, as it is known that the polluted episodes tend to occur with a
weak surface wind and stable boundary layer stratification, which are unfavorable for transport (Zhu et al., 2016;
Tang et al., 2015). Given these uncertainties, there is an urgent need to investigate and determine the common
patterns of ABL structure influence on Beijing's air quality.

On the other hand, the long-term radiosondes are not being fully utilized to investigate urban pollution issues.
The advantage of radiosondes over the other approaches seems to be their length, which usually spans several
decades. For a long time, it was challenging to reduce the wealth of radiosonde datasets to characterize the ABL
structure, therefore, radiosondes remain in very limited use in case studies (Ji et al., 2012; Zhao et al., 2013; Gao
et al., 2016). Recently, self-organizing maps (SOMs; a feather-extracting technique based on an unsupervised
machine learning algorithm) (Kohonen, 2001) were introduced to investigate the ABL thermodynamic structure,
indicating the capabilities of SOMs in feather extraction from a large dataset of the ABL measurements (Katurji et
al., 2015). In fact, the SOM has become increasingly popular in atmospheric and environmental sciences during
the past several years (Jensen et al., 2012; Jiang et al., 2017; Gibson et al., 2016; Pearce et al., 2014; Stauffer et al.,
2016), including a first application of routine radiosondes in South Africa (Dyson, 2015). However, there is thus
far no SOM application in pollution-related ABL structure research. It is expected that such a new analytical
approach can tap the potential of routine radiosondes to better understand urban air pollution.

In this study, a long-term analysis regarding the influence of ABL structure on Beijing's air quality is performed
based on the application of SOMs to 4 years (2013-2016) of radiosonde measurements. The SOM is first used to
classify the vertical temperature profiles for identifying predominant ABL types (see section 3.1). A selection of
climatological observations is then subdivided according to the SOM-based ABL classification (see section 3.2).
Finally, we provide a visual insight into air quality variations (including surface pollutant concentrations and
columnar aerosol properties) under various ABL conditions and discuss the potential physical mechanisms behind
their relationships (see section 3.3–3.5). It is expected that such an association between air quality and ABL type
could provide local policy makers with useful information for improving the predictions of urban air quality.

**2 Materials and methods**
2.1 Data preparation and preprocessing
Radiosonde data observed at the Beijing Observatory (39.81 $\overset{\circ}{}$N, 116.48 $\overset{\circ}{}$E, WMO station number 54511) were
collected from the University of Wyoming (http://weather.uwyo.edu/). The data cover the recent 4-year period





from 2013 to 2016. The Beijing Observatory launches a routine radiosonde twice a day (08:00 and 20:00 Beijing
Time (BJT), corresponding to the morning and evening, respectively) and provides atmospheric sounding data
(profiles of temperature, relative humidity, wind speed, etc.) at the mandatory pressure levels (e.g., surface, 1000,
925, 850, 700 hPa) and additional significant levels. In addition, the hourly near-surface meteorological
parameters (including temperature, wind speed and relative humidity) are also collected from the Beijing
Meteorological Bureau.

We chose the 2000 m above ground level (AGL) as the upper limit of the ABL based on a number studies
investigating the ABL height over Beijing or North China (Tang et al., 2016; Guo et al., 2016; Miao et al., 2017).
This height exceeds the ABL height in most cases, and therefore, most ABL processes influencing the near-surface
air quality are included in the analysis herein. We classify the daily ABL types using the SOM algorithm. To keep
a whole night, the daily vertical profiles are composited from the radiosonde measurements at 20:00 and 08:00 of
the next day.

The mass concentrations of atmospheric pollutants (including $SO_2$, $NO_2$, CO, $O_3$, $PM_{10}$ and $PM_{2.5}$) over Beijing
during the period from 2013 to 2016 are obtained from the Ministry of Environmental Protection of the People's
Republic of China (http://datacenter.mep.gov.cn/). In addition, hourly $PM_{2.5}$ measured at the Beijing US Embassy
(http://www.stateair.net/) are also used in this study. Hourly concentrations are calculated for the Beijing urban
area by averaging concentrations from nine urban sites (including Dongsi, Guanyuan, Tiantan, Wanshouxigong,
Aotizhongxin, Nongzhanguan, Gucheng, Haidianwanliu and US Embassy). To maintain consistency with ABL
classification, the daily pollutant concentration is performed from noon-to-noon (12:00 h–12:00 h).

In addition to near-surface observations, columnar aerosol parameters (including aerosol optical depth (AOD),
Ångström exponent (AE), single scattering albedo (SSA), volume particle size distribution (d$V$/dln$R$), aerosol
radiative forcing (ARF) and so on) are also collected from the AERONET Beijing (39.98°N, 116.38°E) and
Beijing-CAMS (39.93°N, 116.32°E) sites. The level-2.0 quality-assured columnar aerosol data from 2013 to 2016
are downloaded from the AERONET data archive (http://aeronet.gsfc.nana.gov). The size distribution is retrieved
in 22 logarithmically equidistant bins in a range of sizes from 0.05 to 15 μm through a combined spherical and
spheroid particle model (Dubovik and King, 2000; Dubovik et al., 2006).

2.2 Self-organizing maps technique





The SOM is an ideal tool for feather extraction because the input data are treated as a continuum without
relying on correlation, cluster or eigenfunction analysis (Liu et al., 2006). Since Kohonen (1982) first proposed
SOM, it has been widely used for data downscaling and visualization in various disciplines (Jensen et al., 2012;
Katurji et al., 2015; Dyson, 2015; Stauffer et al., 2016; Pearce et al., 2014; Jiang et al., 2017). In this study, the
SOM is introduced to classify the ABL structures. We use the code from the MATLAB SOM Toolbox, which is
freely available from http://www.cis.hut.fi/projects/somtoolbox/.

The following provides a simple introduction about the SOM algorithm, and the details can be found in
Kohonen (2001). SOM training is an unsupervised, iterative procedure, and the result is a matrix of nodes (i.e.,
types) that represent the input data. To learn from the input data, every SOM node has a parametric reference
vector with which it is associated, and these reference vectors are randomly generated. After initialization of the
reference vectors, a stochastic input vector is compared to every reference vector, and the closest match, named
the best-matching unit, is determined by the smallest Euclidean distance. Each reference vector is then updated so
that the best-matching unit and its neighbors become more like the input vector. Whether or not a reference vector
learns from the input vector is determined by the neighborhood function. Only reference vectors that are
topologically close enough to the best-matching unit will be updated according to the SOM learning algorithm.

To exclude the influence of actual temperature values, temperature deviation profiles, which are determined by
subtracting the mean temperature of each profile from each level in the profile, are used as the SOM input in this
study. The first step of SOM training is to determine a matrix size of nodes for initializing the reference vectors.
This step is performed subjectively and depends on the degree of generation required (Lennard and Hegerl, 2015).
We test several SOM matrixes and finally select a $3 \times 3$ matrix, because it captured unique profiles without the
profiles being too general as with a smaller matrix or being too similar as with a larger matrix. The batch mode is
chosen to execute the SOM algorithm. This mode is much more computationally efficient compared to the
sequence mode. The other user-defined settings in the SOM software are set at the default, such as the hexagon
topology, Gaussian neighborhood function, etc.

**3 Results and discussion**
3.1 Self-organized ABL types
We constructed a $3 \times 3$ SOM matrix for daily temperature deviation profiles, and the SOM output shown in Fig.
1 represents nine ABL types (i.e., SOM nodes). In Fig. 1, the SOM nodes are plotted in red, and the individual



profiles corresponding to the SOM node are plotted in blue. For comparison, the mean and 25th and 75th
percentiles for the entire period are plotted in cyan. On the SOM plane, the most notable feather is adjacency of
like types (e.g., nodes 1 and 2) and the separation of contrasting types (e.g., nodes 1 and 9). Although the SOM
nodes appear to be sorted in a certain order, there is no physical significance associated with this order. Such
ordering is a feather of the SOM algorithm (i.e., 'self-organized'). This feather allows us to visualize subtle
differences between the neighboring clusters of profiles and distinguish the unique characteristics of nodes
through the variation of specific features across the SOM plane.

The SOM classification reveals that for the whole study period, the ABL is dominated by near neutral to strong
stable conditions, as none of the SOM nodes fall within the unstable category (i.e., super-adiabatic condition). The
results are reasonable, considering the daily temperature profile is composited from 20:00 and 08:00
measurements. According to the SOM ordering feather, the SOM nodes in four corners (i.e., nodes 1, 3, 7 and 9)
can be thought of as the typical ABL types and the others can be considered transitional ABL types. It is clear
from the individual profiles in Fig. 1 that node 1 represents the well-mixed (near-neutral) condition with no
temperature inversion, node 3 indicates the ABL type dominated by elevated inversion, node 7 indicates the ABL
type dominated by surface inversion, and node 9 represents the ABL type associated with multiple inversions (i.e.,
including surface and elevated inversions).

Frequency analysis of the nine ABL types indicates that the frequency distribution across the types is quite
varied from the expected 11.1 %, with the occurrence frequency showing a 5:1 range from the most frequent type
(node 1) to the least frequent type (node 5). The higher-frequency types are presented on the outer portions of the
SOM plane, while lesser-frequency types are presented closer towards the center (top-right in Fig. 1). The most
dominant types are nodes 1 and 3, and their occurrence frequencies reach 22 % and 20 %, respectively. As
synoptic circulations change with the seasons over Beijing, the ABL types are expected to correspond to
seasonality. The number of profiles from each season in each ABL type is expressed as a percentage and is shown
in Fig. 2. All of the types exhibit strong seasonality. For example, node 1 has the highest occurrence in spring
(29.4 %) and the lowest occurrence in autumn (13.7 %); node 9 presents the highest occurrence in winter (16.3 %)
and the lowest occurrence in summer (4.9 %).

3.2 Evaluation against meteorological data
Fig. 3 shows the average vertical profiles of potential temperature, wind speed and relative humidity



corresponding to each ABL type. As seen in Fig. 3, each of the ABL types is associated with distinct dynamic and
thermodynamic conditions. The potential temperature profiles vary from near neutral conditions to strong stable
conditions, and this change is closely related to the variance in wind speed, suggesting a strong coupling between
the dynamic and thermal effects. The two extreme types (nodes 1 and 9) provide a very useful example. Node 9 is
a very strong stable profile, and the wind speeds are very low in the lower ABL. In contrast, node 1 is a
well-mixed (near neutral) profile and it corresponds to significantly higher wind speeds throughout the ABL. In
addition, when the stability of the atmosphere is strong, vertical mixing is suppressed and winds in the lower ABL
become decoupled from the generally stronger wind aloft. This allows moisture, fogs, low clouds and other scalars
to build up within the stable layer. As a result, the stable ABL types usually correspond to high RH in the lower
ABL.

The near-surface meteorological variables are also examined for each of the ABL types. Fig. 4 shows the
diurnal composite plots of surface temperature, wind speed and relative humidity in the four typical ABL types.
As expected, these near-surface variables respond well to the changing ABL structure. Wind speeds are the highest
on the days corresponding to near neutral conditions (i.e., node 1). High wind speeds result in a deep,
mechanically mixed layer, and these days also exhibited the smallest diurnal amplitude in wind speed, temperature
and relative humidity. Such characteristics are likely consistent with the passage of frontal systems. In contrast,
the smallest wind speeds are observed on days related to strong stable conditions (i.e., node 9). The stable days
also generally exhibit the greatest amplitude of the diurnal signals in temperature and relative humidity. This fact
is an indication that stable conditions occur mostly on clear sky days.

3.3 Evaluation against surface air quality
The concentrations of gaseous and particulate pollutants in the atmosphere are governed by the rate at which
they are emitted from their respective sources, lost by various sink mechanisms, and characteristics of the
atmospheric volume into which they mix. While the mixing volume is determined primarily by the boundary layer
structure, the chemical transformation also depends on boundary layer meteorology in some cases. In the previous
section, it was seen that the SOM technique is an effective tool for classifying boundary layer structures. In this
section, we used the classification technique to quantify the influence of the boundary layer structure on
near-surface air quality.

Fig. 5 examines the daily concentrations of gaseous and particulate pollutants in relation to various ABL types.




As expected, the most stable conditions are associated with a dramatic increase in the mass concentrations of air
pollutants (except $O_3$). On average, $SO_2$, $NO_2$, CO, $PM_{10}$ and $PM_{2.5}$ increase by 15.7 µg/m$^3$ (142 %), 44.3 µg/m$^3$
(119 %), 1.5 mg/m$^3$ (202 %), 91.6 µg/m$^3$ (119 %) and 95.9 µg/m$^3$ (218 %) from the near neutral ABL condition
(i.e., node 1) to strong stable condition (i.e., node 9), respectively. The highest increasing amplitude is related to
$PM_{2.5}$, suggesting fine particulate matters are likely accumulated from not only primary emissions but also
secondary formation (Zhang and Cao, 2015). As we have shown, the more stable ABL conditions tend to
correspond to high relative humidity in the lower ABL (Figs. 3 and 4). Additional enhancement in $PM_{2.5}$ can be
expected under the humid condition, as it is known that the humidity-related physicochemical formation of
particles (such as hygroscopic growth, liquid-phase and heterogeneous reactions) can be intensified by high
humidity values (Cheng et al., 2015; Cheng et al., 2016; Zheng et al., 2015).

Interestingly, increasing atmospheric stability has an opposite effect on near-surface $O_3$ concentrations. Since
$O_3$ is produced by photochemical interactions between $NO_x$ (NO + $NO_2$) and volatile organic compounds (VOCs)
(Seinfeld and Pandis, 2006), the boundary layer structure alters the $O_3$ level through modulation of its precursors
($NO_x$ and VOCs). The low $O_3$ level in the stable ABL can be explained by the strong titration reaction. Since $O_3$ is
highly reactive, when trapped in a stable layer, surface titration by the NO emitted from vehicles can cause a rapid
reduction in $O_3$ concentration. In previous studies, persistent low $O_3$ concentration were observed in the stable
boundary layer condition in Beijing (Zhao et al., 2013). Conversely, when near-surface wind speeds are higher
(near neutral condition such as node 1), $O_3$ is mixed downward from the overlying air mass, resulting in higher
concentrations. Nevertheless, it is worth noting that the extremely high $O_3$ values (not shown) were also detected
on very stable days (i.e., node 9), suggesting the complexity of $O_3$ behavior in response to the boundary layer
structure (Tong et al., 2011; Haman et al., 2014).

To obtain a more in-depth understanding of the physical mechanisms behind the relationship between air
quality and ABL structure, diurnal composite hourly concentrations of atmospheric pollutants are formed for each
ABL type. The SOM-based ABL classification scheme provides a consistent, gradual distinction in the diurnal
cycles of surface air pollutants from near neutral to strong stable conditions. The composite diurnal evolutions of
air pollutants in the four typical ABL types (i.e., nodes 1, 3, 7 and 9) are illustrated in Fig. 6. The diurnal cycles of
$SO_2$, $NO_2$, CO, $PM_{10}$ and $PM_{2.5}$ are extremely pronounced under the strong stable condition (i.e., node 9),
although very reduced under the near neutral condition (i.e., node 1). In contrast, the behavior of $O_3$ is completely
different from other pollutants. The results suggest that the chemical species, which are mainly produced by





surface emissions, are strongly modulated by the development of the ABL, while the chemical species, which are
strongly controlled by the photochemical process, are weakly regulated by the development of the ABL (Crawford
et al., 2016). Overall, the diurnal behavior of each pollutant species in each of the ABL types is generally
consistent with the existing knowledge for urban areas (Chambers et al., 2015b; Chambers et al., 2015a; Zhang et
al., 2012b; Jenner and Abiodun, 2013; Han et al., 2009).

Of particular interest in Fig. 6 is that (1) nodes 3 and 9 have similar magnitudes of concentrations in the
afternoon, and (2) nodes 7 and 9 have similar increments in concentrations from afternoon to midnight, although
there is a huge distinction in the afternoon concentrations (i.e., afternoon baselines). This sheds some light on the
common patterns of the ABL controls on the near-surface air quality in Beijing. Considering the thermal inversion
feather in each of ABL types (Fig. 1), the regulating effects of ABL on near-surface concentrations can be
concluded as follows: (1) elevated inversion enhances the afternoon baseline; and (2) surface inversion improves
the evening increment. Obviously, the high afternoon baselines in nodes 3 and 9 can be attributed to elevated
inversion, while high evening increments in nodes 7 and 9 can be attributed to surface inversion. Since surface
inversion usually develops shortly before sunset due to radiation cooling, the evening traffic emission peak is
counteracted by a stabilizing boundary layer. Consequently, the air pollutants such as $NO_2$, $CO$, $PM_{10}$ and $PM_{2.5}$
often experience an explosive growth from afternoon to midnight. In contrast, elevated inversion usually forms
due to synoptic forcing (such as synoptic advection) (Hu et al., 2014; Xu et al., 2016) and can persist for several
days; as a result, the daytime mixing volume is also depressed, causing a relatively higher afternoon
concentration.

Beijing has relatively little industry but numerous automobiles, and the emissions of $SO_2$ are small while those
of $CO$, $NO_x$ and particles are much larger (Zhao et al., 2012). By comparison, the diurnal behaviors of $SO_2$ and
other pollutants are completely different. For example, in node 9, $SO_2$ show a lower nighttime concentration but a
sharp increase after sunrise, whereas $NO_2$, $PM_{10}$ and $PM_{2.5}$ show a higher nighttime concentration with a slight
morning increase associated with the traffic emission. The results largely suggest that the changing ABL structure
affects the near-surface observations of locally and remotely sourced pollutants in very different ways. In the
evening, since the stable boundary layer (SBL) and the residual layer (RL) are essentially decoupled with each
other (Stull, 1988), locally sourced pollutants emitted into the surface layer (such as $CO$, $NO_2$ and particulate
matters from vehicular emissions) become trapped close to the surface. In contrast, remotely sourced pollutants
emitted from chimneystacks above the SBL (such as $SO_2$ from power plants in the Hebei Province) may be stored





within the RL aloft and not penetrate into the SBL. As the daytime convective turbulent mixing developed in the
morning, the rapid momentum transfer between the surface and aloft air transported the pollutants stored in RL
downward and meanwhile upwardly mixed the pollutants trapped from the previous night in the surface layer
(Salmond and McKendry, 2005). It is observed in Fig. 6 that after a stable night, the burst of turbulent activity in
the morning coincides with a rapid increase in $SO_2$ concentration (Fig. 6). Since there is no significant increase in
$SO_2$ emission at the surface at this time, this result strongly suggests that increased $SO_2$ in the morning resulted
from the downward mixing of stored $SO_2$ in the RL aloft. In a previous case study, Li et al. (2017b) reported that
as a result of both turbulent mixing and the advection of high concentrations of air pollutants above the surface
layer, the urban area of Beijing experienced a dramatic increase of the $PM_{2.5}$ concentration in the morning on 30
November 2015.

Given the importance of local vehicle emissions vs. more-distance power plant and industrial emissions for
Beijing's air quality, the ratio of $CO/SO_2$ can be considered as an indicator of the contribution of local emissions
to air pollution, with higher ratios indicating higher local contributions (Tang et al., 2015). Fig. 7 shows the
composite diurnal variations of $CO/SO_2$ ratios in the four typical ABL types (i.e., nodes 1, 3, 7 and 9). The
contrasts between $CO/SO_2$ ratios for the various ABL types are noticeable during the nighttime, whereas
differences during the daytime are minimal. During the daytime, when the ABL is well mixed, near-surface
pollutant concentrations represent a combination of local and remote sources. In the evening, however, the earth's
surface begins to cool, and a stable boundary layer begins to form from the ground up. If sufficiently strong, the
nocturnal surface inversion can isolate near-surface observations from the influence of distant sources (Crawford
et al., 2016). Consequently, the more stable the nocturnal conditions near the ground, the higher the $CO/SO_2$ ratios
that occur (Fig. 7). The results are consistent with previous studies (Tang et al., 2015; Zhu et al., 2016), indicating
local contribution increases with enhanced static stability in the surface layer over Beijing. According to the above
analysis, high pollutant loadings in node 9 are mostly attributable to local contributions (the highest $CO/SO_2$ ratios
in node 9); however, high pollutant loadings in node 3 are more likely due to regional contributions (the lowest
$CO/SO_2$ ratios in node 3). Obviously, it is the stable stratification rather than the weak surface wind that confines
the regional contribution.

3.4 Evaluation against columnar aerosol pollution
For many years, aerosol particles have been the primary pollution problem in Beijing. Atmospheric aerosols
play an important role in radiation transfer due to absorption and/or scattering in the atmosphere, and thus could





have a great influence on the evolution of the ABL. In recent years, the feedback effect of aerosols on the ABL has
drawn much attention (Kajino et al., 2017; Gao et al., 2016; Ding et al., 2016). To further our understanding of
aerosol pollution in Beijing, we examine the optical and physical properties and the direct radiative forcing of
columnar aerosols in the different ABL types in this section.

Aerosol optical properties can be characterized by three useful parameters: AOD, AE and SSA. Fig. 8 illustrates
the $AOD_{440nm}$, $AE_{440nm-870nm}$ and $SSA_{440nm}$ over Beijing within the nine ABL types. The ABL-type averages of
AOD range from 0.52 and 1.22 (Fig. 8a). Comparing with near-surface observations, the greatest difference is that
the highest AOD value generally occurs in node 3, rather than in node 9 (the highest surface $PM_{2.5}$ and $PM_{10}$
concentrations occur in node 9). This may be attributed to the difference in aerosol vertical distribution in these
two types. As we have demonstrated in Sect 3.3, node 3 is related to strong regional transport. Since the height of
regional transport is usually above the surface layer, such as 200–700 m AGL detected by Li et al. (2017a), more
aerosol particles might be suspended above the surface layer in node 3, resulting in the highest AOD value in the
atmospheric column. In addition, since high relative humidity also occurs in node 3, the highest AOD value in this
ABL type could be partly attributed to the particle hygroscopic growth (Chen et al., 2014; Deng et al., 2016; Zhao
et al., 2017).

It is known that high AE values indicate a dominance of fine particles, while low values indicate a dominance
of coarse particles. Unlike AOD, the AE shows a relatively low sensitivity to ABL types (Fig. 8b). All type
averages of AE are higher than 1.0, suggesting that the proportion of fine particles is always larger than that of
coarse particles over Beijing (Yu et al., 2017; Yu et al., 2009). The highest AE occurs in node 6 (1.20) and the
lowest is 1.03 in node 1. Node 1 corresponds to the lowest AE value, indicating that under the near neutral ABL
condition, the coarse particles contribute a relatively higher proportion of total particles. This could be due to the
increasing wind speed with decreasing relative humidity (Figs. 3 and 4). Coarse particles could be from more
natural and anthropogenic dust emission under high wind speed conditions. Particularly during the fast
northwesterly wind period, dust storms occasionally contribute to the high coarse particle loadings in Beijing (Yu
et al., 2016). The long-distance transport of dust particles from northwest China may be the reason for the lowest
AE value in node 1.

The SSA is defined as the ratio of the scattering coefficient and the total extinction coefficient. It is mostly
dependent on the aerosol size, concentration of absorbing component and its mixture state with non-absorbing



components. The daily SSA at 440 nm ranges from 0.82 to 0.98 during the study period, which suggests that there
are quite different types of aerosols in the columnar atmosphere over Beijing (varying from strong absorbing
aerosols to strong scattering aerosols). It is easy to see that the ABL types associated with a strong surface
inversion (i.e., nodes 7, 8 and 9) have lower SSA values (Fig. 8c). The averaged SSA in these nodes is
approximately 0.90, which is significantly lower than that in nodes 1, 2 and 3. The low SSA values mean
enhancement in the absorbing particles, such as black carbon, which are released from industry, biomass/biofuel
burning, diesel vehicle, and coal burning. In contrast, the highest SSA occurring in node 1 can be explained by
dust particle transmission and soil aerosol emissions.

351        The volume particle size distribution retrieved in the sizes between 0.05 and 15 μm is one of the most important

parameters for studying the behavior of aerosols (Dubovik and King, 2000). Fig. 9 expresses the mean volume
particle size distribution ($dV/d\ln R$) over Beijing in the nine ABL types. Table 1 supplements Fig. 9 with the
statistical parameters of aerosol particle size distribution. Clearly, the volume particle size agrees very well with
the bimodal lognormal distributions. Both fine ($R < 0.6$ μm) and coarse ($R > 0.6$ μm) modes exhibit relative
stability with two peaks at a radius of approximately 0.1–0.2 μm and 2.0–4.0 μm, which are similar to some
previous studies (Eck et al., 2005; Xia et al., 2007; Che et al., 2014). However, the size distribution shows a
distinct difference in the changing amplitude for different ABL types. The fine- and coarse-mode particle volumes
increase rapidly from left (nodes 1, 4 and 7) to right (nodes 3, 6 and 9) on the SOM plane. This suggests that with
the stabilizing boundary layer processes, the atmosphere is more loaded with both fine- and coarse-mode particles
over Beijing. In addition, the stabilizing processes are accompanied by the increase of the fine-mode effective
radius ($R_{eff}$-F) and fine-mode volume fraction ($Vf/Vt$). These results strongly point to the important role of
fine-mode particle hygroscopic growth on the days associated with stable nocturnal ABL conditions.

365        The type-averaged ARF at the surface (BOA), top of atmosphere (TOA), and within the atmosphere (ATM)

over Beijing is shown in Fig. 10. The type averages of ARF at the surface range from -47.8 W/m$^2$ to -110.0 W/m$^2$,
while at the TOA, they are found to be between -21.1 W/m$^2$ and -48.0 W/m$^2$. Likewise, the ABL type averaged
ARF within the atmosphere are between 26.7 W/m$^2$ and 63.1 W/m$^2$. The larger negative ARF at the surface (> 110
W/m$^2$) and positive ARF within the atmosphere (> 60 W/m$^2$) are found in ABL types 3, 6 and 9 over Beijing,
implying strong cooling at the surface and warming in the atmosphere. These results are induced by relatively
larger aerosol loadings under the stagnant meteorological conditions. The larger ARF within the atmosphere
demonstrates that solar radiation is being absorbed within the atmosphere, and as a result, heats the atmosphere





and reduces surface temperature. This can change the atmospheric vertical temperature gradient and improve the
atmospheric stability (Li et al., 2010; Ge et al., 2010; Zou et al., 2017). Finally, the enhanced stability hinders the
vertical diffusion of aerosol particles, leading to the increase of aerosol concentrations and causing a further
decrease in the solar radiation and ABL height, which induces a positive feedback loop for causing high surface
aerosol concentrations (Quan et al., 2013; Zhong et al., 2017).

3.5 Evaluation against heavy polluted episodes
In January of 2013, December of 2015, and December of 2016, heavy aerosol pollution episodes frequently
wreaked havoc across Beijing and its surroundings, which resulted in severe damages to the environment and
human health. Fig. 11 shows the hourly variations of $PM_{2.5}$ and $AOD_{440nm}$ during the three heavily polluted
months. It is observed that the $PM_{2.5}$ concentrations were frequently elevated to above 200 $\mu g/m^3$, and the AOD
often exceeded 1.0 in Beijing during these three months. The ABL types (shown at the top of each plot) reveal that
pollution episodes were generally associated with the control of nodes 3 and 9, and clean episodes were often
associated with the control of node 1. For example, the severe pollution episode that occurred from 9–14 January
2013 was due to the alternate control of nodes 3 and 9, and the pollution episode from 15–21 December 2016 was
related to the persistent control of node 9. In contrast, multiday control of node 1 caused a clean episode from 14–
16 December 2015. The linkages between air quality and the boundary layer structure were consistent with the
long-term analyses described in Sects. 3.3 and 3.4, indicating that the ABL types are one of the primary drivers of
day-to-day variations in air quality over Beijing.

The monthly $PM_{2.5}$ concentrations in the Beijing urban area reached up to 180.8 $\mu g/m^3$, 153.9 $\mu g/m^3$ and 147.9
$\mu g/m^3$ in January 2013, December 2015 and December 2016, respectively. All these values were far larger than the
4-yr winter mean $PM_{2.5}$ concentration (110.6 $\mu g/m^3$). Although the characteristics of $PM_{2.5}$ air quality depend on
many complex elements, the major contributors are the pollutant emissions and meteorological conditions. In
2013, the Chinese State Council released the "Atmospheric Pollution Prevention and Control Action Plan" to
implement a megacity cluster-scale joint prevention and control strategy program. As a result, the $PM_{2.5}$ in Beijing
decreased from 89.5 $\mu g/m^3$ in 2013 to 73.0 $\mu g/m^3$ in 2016. However, these meteorology-driven pollution episodes
to some degree obscure the true impacts of the emission control strategies implemented by government. Fig. 12
shows a comparison of the occurrence frequency of the nine ABL types to the winter mean frequency (2013-2016)
for the three polluted months. Compared with the 4-yr winter mean frequency, the greatest differences are that the
occurrences of nodes 3 and 9 (the two most polluted types) increased and node 1 (the clean type) decreased during



the three polluted months. Obviously, the elevated PM$_{2.5}$ concentrations in the abovementioned months can be
mostly attributable to the anomalous boundary layer structures.

Quantitative analysis of the roles of the ABL anomaly in PM$_{2.5}$ variations during the pollution months is helpful
for the assessment of air pollution prevention and control strategies. In this study, the ABL classification allows
for the integrated evaluation of the effects of numerous interrelated ABL meteorological parameters on air quality.
Here, a meteorology-to-environment method (revised from the circulation-to-environment method proposed by
Zhang et al. (2012a)) is utilized to evaluate the influence of the ABL anomaly for enhanced PM$_{2.5}$ levels during the
abovementioned months. We assume the linkages between ABL types and their PM$_{2.5}$ loadings in winter are
constant in different years. For each polluted month, the total anomaly ($C'$) is defined as the deviation in PM$_{2.5}$
from the 4-yr winter mean concentration ($\overline{C}$). This total anomaly in each month is due to the combined effects of
meteorology and emission. The anomaly calculated from the mean PM$_{2.5}$ loadings for nine ABL types and their
occurrence frequencies during each month can be considered to represent the PM$_{2.5}$ change caused by the
anomalous boundary layer structure. We refer to this as the "ABL-driven" anomaly. The ABL-driven anomaly
($C_{ABL}{}'$) is calculated through $\sum_i F_i \cdot C_i - \overline{C}$ , where $F_i$ is the occurrence frequency of type-$i$ ABL during a
specific period and $C_i$ is the corresponding PM$_{2.5}$ loading feathering that type. The ratio of $C_{ABL}{}'$ to $C'$ (the
difference of $C_{ABL}{}'$ to $\overline{C}$) is then used to evaluate the relative (absolute) contribution of the ABL anomaly to the
enhanced PM$_{2.5}$ level. The results show that the contributions of the frequency anomaly of the ABL type to the
increase in PM$_{2.5}$ are 65.8 % (46.2 μg/m$^3$) during January 2013, 46.7 % (20.2 μg/m$^3$) during December 2015 and
94.6 % (35.3 μg/m$^3$) during December 2016. These quantitative estimations suggest that the ABL anomaly to a
large extent explains the enhanced PM$_{2.5}$ concentrations during these polluted months.

**4. Summary**
The influence of the ABL structure on Beijing's air quality is still unclear due to the lack of long-term
observations. On the other hand, the long years of routine radiosondes remain underutilized as a tool for urban
pollution studies. In this study, the SOM was applied to 4-yr radiosondes to classify the ABL types over Beijing.
The resulting types were then evaluated in relation to meteorological and environmental variables, with an attempt
to understand the roles of different ABL conditions in regulating the air quality in Beijing. The main findings are
as follows:
1)  The SOM provides a continuum of nine ABL types (i.e., SOM nodes), and each type is characterized with



distinct boundary layer meteorological conditions (including dynamic and thermodynamic conditions).
2)  From the near neutral (i.e., node 1) to strong stable ABL types (i.e., node 9), the surface concentrations of $SO_2$,
$NO_2$, CO, $PM_{10}$ and $PM_{2.5}$ on average increase approximately 120–220 %. The diurnal evolutions of these
pollutants are strongly modulated by temperature inversions. While an elevated inversion enhances the
afternoon baseline concentration, the surface inversion improves the evening concentration increment. In
contrast, $O_3$ show an opposite variation in response to the ABL types.
3)  Boundary layer evolution affects the near-surface observations of locally and remotely sourced pollutants in
very different ways, causing a distinct difference in the diurnal variations of $SO_2$ and other pollutants (e.g.,
$NO_2$, CO, $PM_{10}$ and $PM_{2.5}$). Comparing the $CO/SO_2$ ratios in different ABL types reveals that the local
contribution increases with enhanced static stability near the ground, and it is the stable boundary layer
stratification rather than weak surface wind that confines the regional contribution.
4)  With the stabilizing ABL processes, the atmosphere column is more loaded with both fine- and coarse-mode
particles. Node 3 (dominated by elevated inversion and high relative humidity) corresponds to the most severe
columnar aerosol pollution, characterized by the highest optical depth (1.22) and volume concentration (0.30
$\mu m^3/\mu m^2$). The larger negative ARF at the surface (> 110 $W/m^2$) and positive ARF within the atmosphere (>
60 $W/m^2$) are associated with the three stable ABL types (i.e., nodes 3, 6 and 9), suggesting the possible
influence of a positive feedback loop for causing high surface aerosol concentrations.
5)  Analysis of three typical pollution months (i.e., January 2013, December 2015 and December 2016) suggests
that the ABL types are one of the primary drivers of day-to-day variations in Beijing's air quality. During the
three pollution months, the frequency of stable ABL types (e.g., nodes 3 and 9) increases significantly
compared with the 4-yr (2013-2016) winter mean frequency. In contrast, the frequency of the well-mixed
ABL type (i.e., node 1) is greatly reduced during these pollution months.
6)  Using a meteorology-to-environment method, the relative (absolute) contribution of the ABL anomaly to
enhanced $PM_{2.5}$ level is estimated to be 65.8 % (46.2 $\mu g/m^3$) during January 2013, 46.7 % (20.2 $\mu g/m^3$) during
December 2015, and 94.6 % (35.3 $\mu g/m^3$) during December 2016.

This work revealed the common pattern of the influence of different ABL structures on Beijing' air quality. The
established correlations between ABL type and air quality could be useful for developing an operational forecast
and warning system. In addition, this work demonstrated that the SOM-based ABL classification scheme is a
powerful tool for understanding urban air pollution. Since the SOM technique is good at feather extraction, the
coarse-resolution radiosonde profiles can be taken as the SOM input (as we have shown in this study). Therefore,





it can take advantage of the long-term available radiosondes, which is simple and economical to implement in
comparison to conventional techniques (such as mooring boats, airplane, and ground remote sensing). We believe
that the pollution-related ABL research and the formulation of pollution control measures could benefit from
application of the SOM analytical tool.

**Data availability**
The datasets used in this study are publicly available at the University of Wyoming (http://weather.uwyo.edu/), the
Ministry of Environmental Protection of the People's Republic of China (http://datacenter.mep.gov.cn/), the U.S.
Department of State Air Quality Monitoring Program (http://www.stateair.net/), and the Aerosol Robotic Network
(https://aeronet.gsfc.nasa.gov/).

**Competing interests**
The authors declare no conflict of interest.

**Acknowledgements**
This study is supported by the National Key Research and Development Plan of China (Nos. 2017YFC0209606
and 2016YFC0203305), the National Natural Science Foundation of China (Nos. 41630422, 41475140 and
41475004) and the Special Fund for Basic Scientific Research Business of Central Public Research Institutes
(PM-zx703-201601-019). The authors would like to thank the Beijing Meteorological Bureau, the Ministry of
Environmental Protection of the People's Republic of China and the Wyoming Weather Web for providing related
data. Concerning the AERONET data used in this paper, we are particularly grateful to Prof. Huizheng Che, Prof.
Hongbin Chen and Prof. Philippe Goloub for their efforts in establishing and maintaining the AERONET site in
Beijing, as well as their assistants for the upkeep of the instrument and availability of the online data.

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





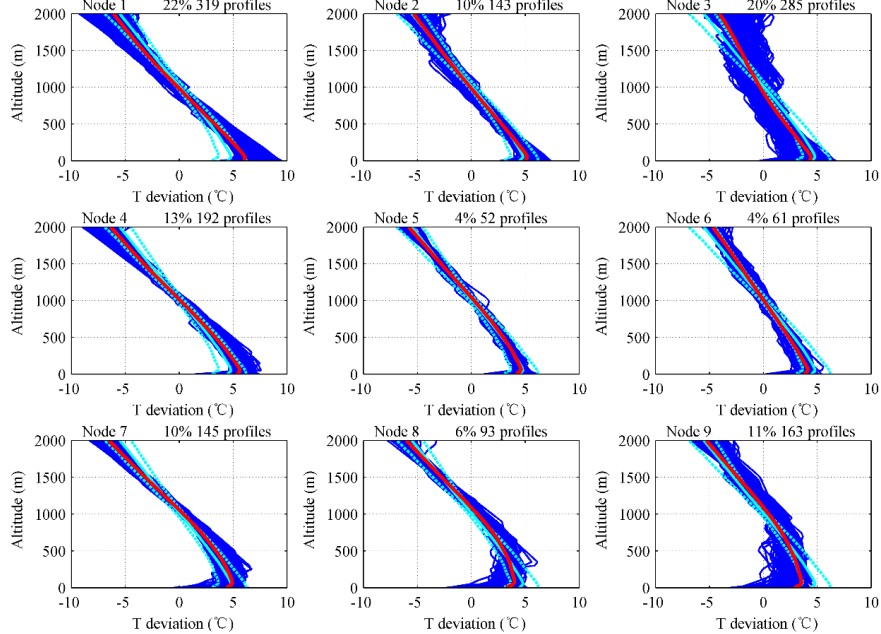

**Figure 1. The 3 × 3 SOM output for radiosonde-based temperature (T) deviation profiles observed at the Beijing Observatory. SOM nodes are shown in red, with the corresponding individual profiles in dark blue. For reference, the overall average temperature profile and 25th and 75th percentile profiles are shown in cyan. The top-right shows the occurrence cases and frequency of each SOM node.**
























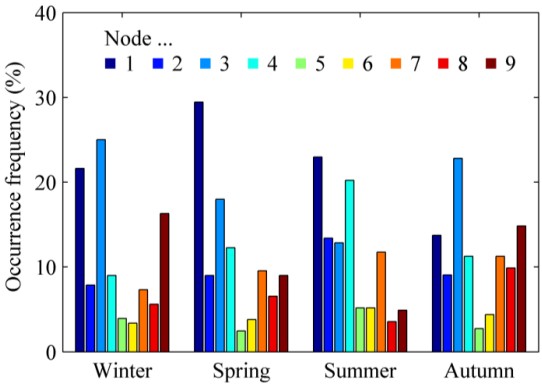


**Figure 2. Seasonality of SOM nodes shown as the relative frequency of seasons within each SOM node.**
**Winter (DJF); Spring (MAM); Summer (JJA); Autumn (SON).**

























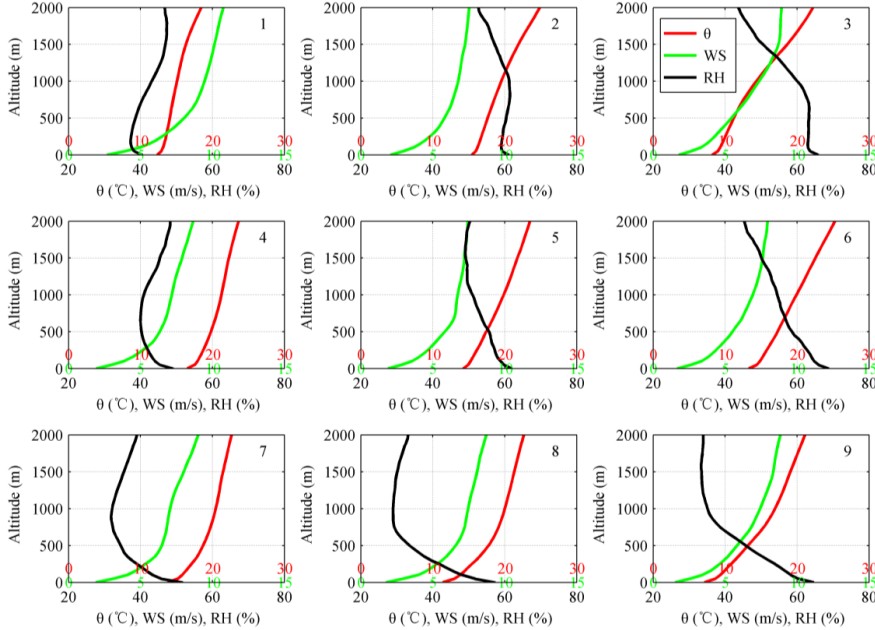


**Figure 3. Profiles of average potential temperature (θ), wind speed (WS) and relative humidity (RH) corresponding to each SOM node at the Beijing Observatory. The red, green and black labels of the horizontal axis correspond to θ, WS and RH, respectively.**


















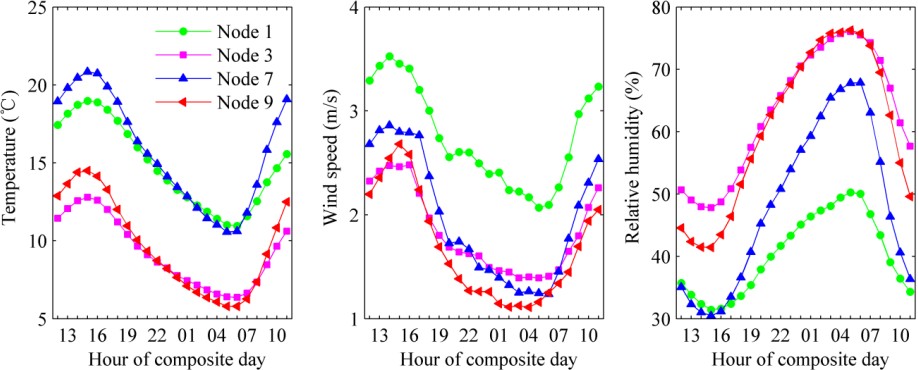


**Figure 4. Hourly mean diurnal composites of temperature, wind speed and relative humidity in Beijing**

**corresponding to SOM nodes 1, 3, 7 and 9.**

























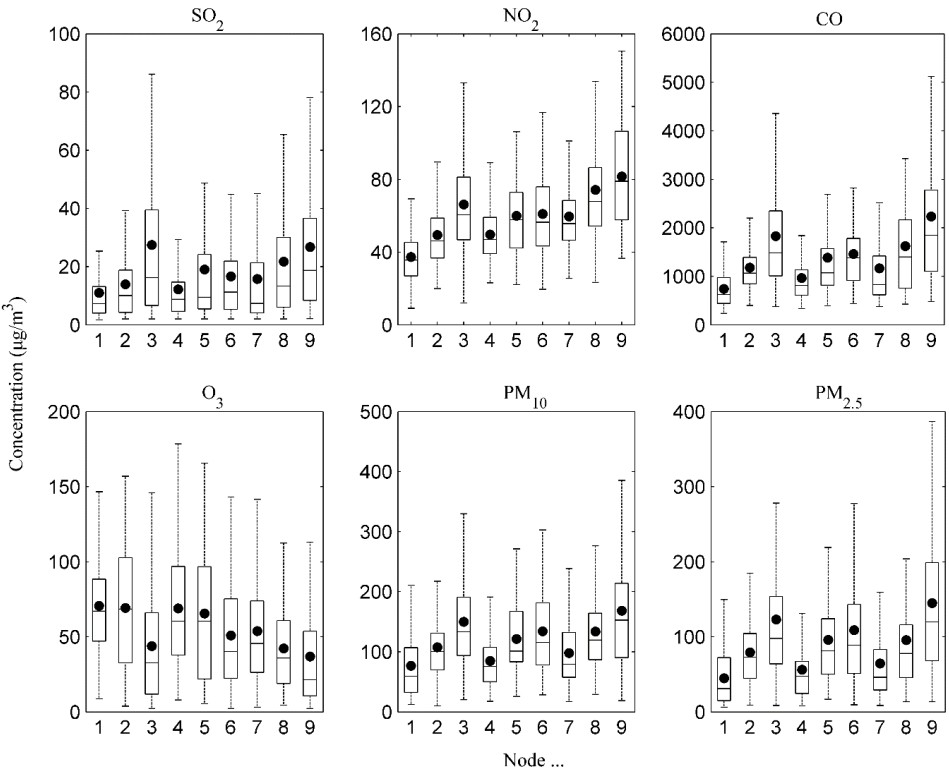


**Figure 5. Daily pollutant concentrations in Beijing corresponding to each SOM node. The solid dots denote the mean. The box and whisker plot presents the median, the first and third quartiles, and the 5th and 95th percentiles, respectively.**




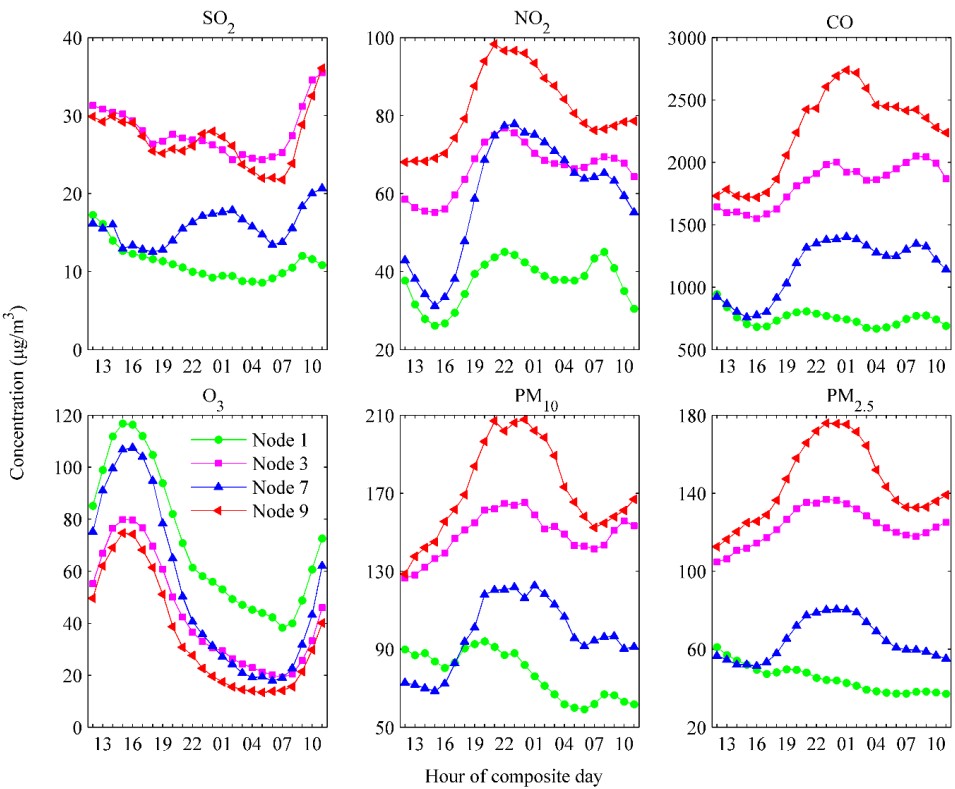


**Figure 6. Composite diurnal variations of air pollutants in Beijing corresponding to SOM nodes 1, 3, 7 and**
**9.**

















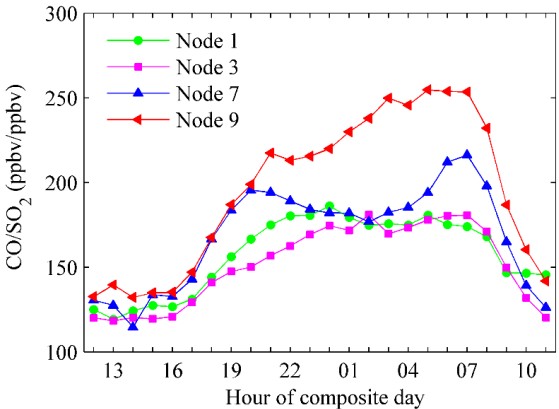


**Figure 7. Composite diurnal variations of CO/SO₂ ratios in Beijing corresponding to SOM nodes 1, 3, 7 and**

**9.**
























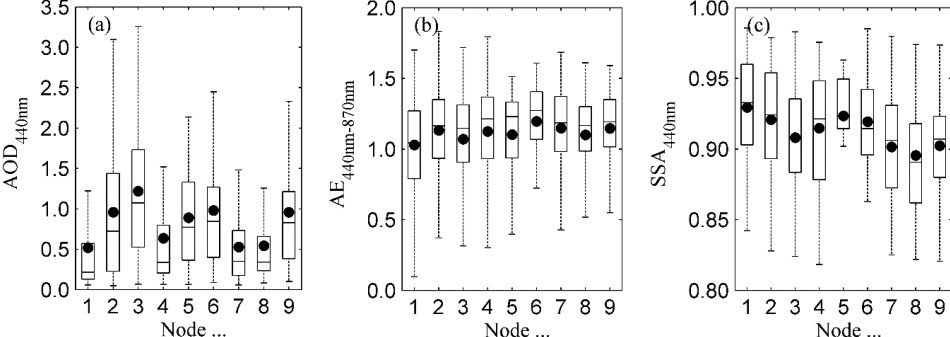


**Figure 8. (a) Aerosol optical depth (AOD$_{440nm}$), (b) Ångström exponent (AE$_{440nm-870nm}$) and (c) single**

**scattering albedo (SSA$_{440nm}$) over Beijing and corresponding to each SOM node. The solid dots denote the**
**mean. The box and whisker plot presents the median, the first and third quartiles, and the 5th and 95th**
**percentiles, respectively.**
























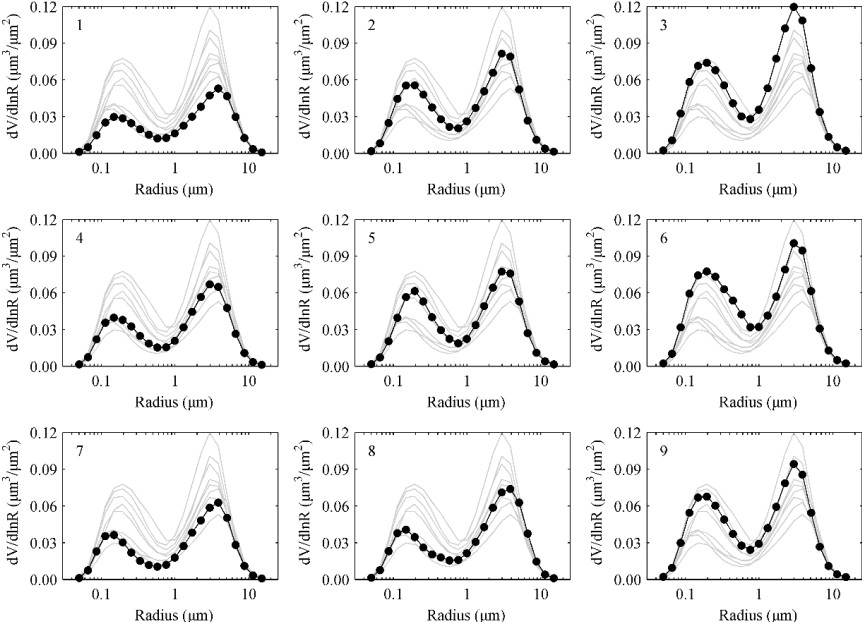

**Figure 9. Mean volume particle size distribution over Beijing corresponding to each SOM node. The average volume particle size distribution for each node is shown by the gray line and is repeated on each plot for comparison. The size distribution for each type is highlighted in the black dotted line on the respective plot.**





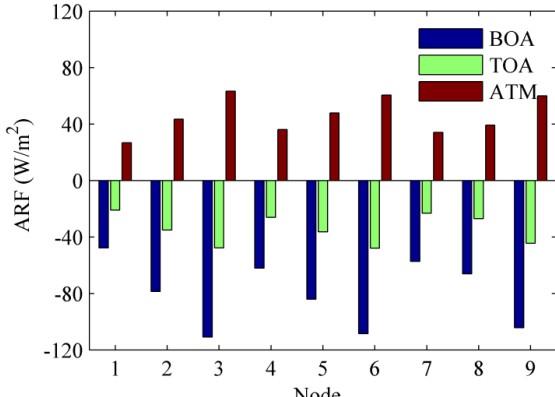


**Figure 10. Aerosol radiative forcing (ARF) at the surface (BOA), top of atmosphere (TOA), and within the**

**atmosphere (ATM) over Beijing and corresponding to each SOM node.**

























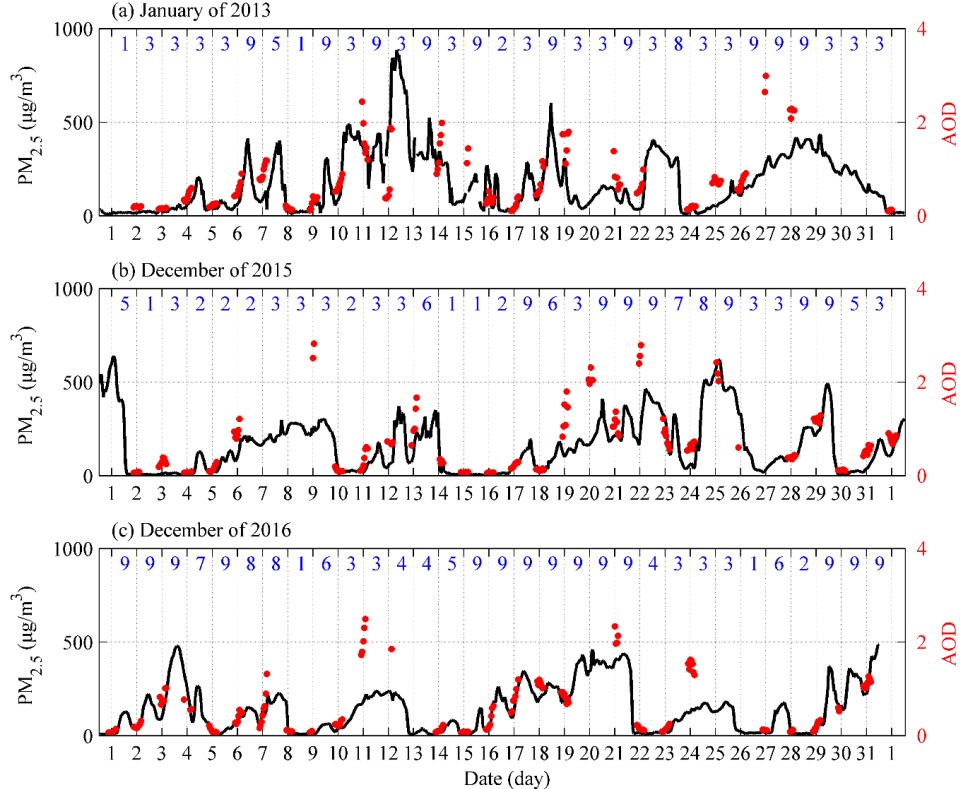


**Figure 11. Time series of hourly PM$_{2.5}$ and AOD$_{440nm}$ in (a) January of 2013, (b) December of 2015, and (c)**


**December of 2016. The daily SOM nodes (i.e., ABL types) are shown at the top of each plot (blue numbers).**















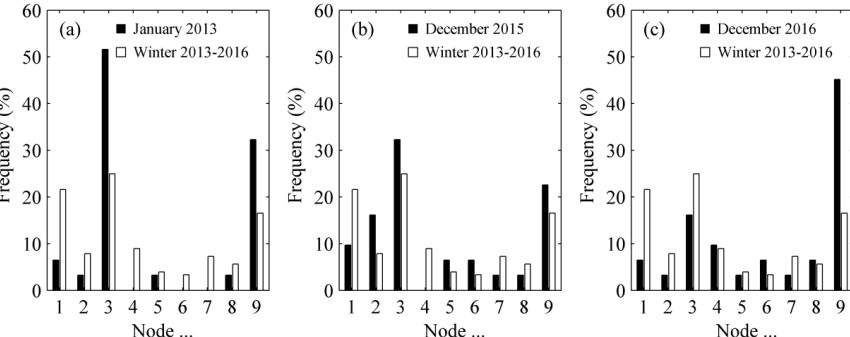


**Figure 12. Occurrence frequency of the SOM nodes during (a) January 2013, (b) December 2015, and (c)**

**December 2016.**





























| Node ... | Effective radius (μm) | | | Volume concentration (μm³/μm²) | | | Vf/Vt |
|---|---|---|---|---|---|---|---|
| | $R_{eff}$-T | $R_{eff}$-F | $R_{eff}$-C | VolCon-T | VolCon-F | VolCon-C | |
| 1 | 0.54 | 0.14 | 2.46 | 0.13 | 0.05 | 0.09 | 0.31 |
| 2 | 0.41 | 0.16 | 2.42 | 0.21 | 0.09 | 0.12 | 0.42 |
| 3 | 0.45 | 0.16 | 2.30 | 0.30 | 0.12 | 0.18 | 0.41 |
| 4 | 0.45 | 0.15 | 2.38 | 0.17 | 0.06 | 0.11 | 0.36 |
| 5 | 0.47 | 0.17 | 2.44 | 0.21 | 0.09 | 0.12 | 0.41 |
| 6 | 0.38 | 0.17 | 2.33 | 0.28 | 0.14 | 0.14 | 0.45 |
| 7 | 0.42 | 0.14 | 2.37 | 0.15 | 0.05 | 0.10 | 0.34 |
| 8 | 0.44 | 0.14 | 2.32 | 0.18 | 0.06 | 0.12 | 0.35 |
| 9 | 0.38 | 0.16 | 2.25 | 0.25 | 0.11 | 0.14 | 0.43 |

**Table 1. Statistical parameters of aerosol particle size distribution corresponding to each SOM node.**
**VolCon is the volume concentration; $R_{eff}$ the effective radius; Vf/Vt denotes the fine-mode volume fraction.**
**T, F, and C represent the total, fine-, and coarse-mode particles.**