# Peer review of "Self-organized classification of boundary layer meteorology and associated"

_Atmospheric Chemistry and Physics, 2017_

## Referee Comment (RC1) · Anonymous Referee #2 · 31 Jan 2018

General comments

The paper deals with the influence of boundary layer structure on air quality using 4-year observations. The article presented very interesting study between air pollutants and meteorology, and the study built on very good meteorological measurements data. The paper is certainly worth of publishing as the study itself is extremely interesting. However, some improvements/corrections are suggested.

Specific comments

1. The title should be modified. Throughout the manuscript, I haven't found the description concerning the calculation of the BLH. Therefore, the title using boundary

layer structure is not exactly correct. Maybe atmospheric stability or boundary layer meteorology are much better. In addition, 4-year is not long-term for the observation period. In section 3.4, the authors also discussed the impact of the aerosols on BLH. However, the title just represented the influence of boundary layer meteorology on air quality. I suggest removing section 3.4 or please revise the title correspondingly.

2. The radiosondes the authors used mainly represent the stable condition of the boundary layer. However, the air pollutants were observed in the whole day. Especially, the AOD data were just observed during the daytime. As well known, the boundary layer height will develop rapidly during the daytime. Lacking of the daytime observations of boundary layer structures, how about the creditability of the relationship of air pollutants and boundary layer meteorology?

3. The instruments of US Embassy for PM are not the same as the MEP. Do you consider the differences?

4. For the boundary layer meteorology, potential virtual temperature and Richardson number are much better indicators to do the SOM analysis. Why the authors using temperature to do the SOM analysis? I suggest using potential virtual temperature to do the classification of different nodes.

5. How to evaluate atmospheric stability? Do you have the quantitative basis? Node 3 represent slow wind, high humidity and more stable than node 7.

6. Because emissions are different in different seasons, I suggest discussing the concentrations of air pollutants in different nodes for each seasons.

7. For the boundary layer ozone analysis, please refer Tang et al., 2017a, 2017b.

Technical comments

1. Line 48. Beijing has two directions adjacent to mountains. The west one is Tai hang Mountains and two north one is Yan Mountains.

[Figure]

2. Line 56. Ground-based remote sensing.

3. The second paragraph in the introduction section is too long. Please separate it.

4. Section 3.3 is too long, please use subtitle to separate it into several small parts.

References

Tang, G., Zhu, X., Xin, J., Hu, B., Song, T., Sun, Y., Zhang, J., Wang, L., Cheng, M., Chao, N., Kong, L., Li, X., and Wang, Y. Modelling study of boundary-layer ozone over northern China - Part I: Ozone budget in summer. Atmos. Res., 187, 128-137, doi: 10.1016/j.atmosres.2016.10.017, 2017a.

Tang, G., Zhu, X., Xin, J., Hu, B., Song, T., Sun, Y., Wang, L., Wu, F., Sun, J., Cheng, M., Chao, N., Li, X., and Wang, Y. Modelling study of boundary-layer ozone over northern China - Part II: Responses to emission reductions during Beijing Olympics. Atmos. Res., 193, 83-93, doi: 10.1016/j.atmosres.2017.02.014, 2017b.

---

## Referee Comment (RC2) · Anonymous Referee #1 · 7 Feb 2018

This manuscript uses an unsupervised machine learning to understand the relationships between boundary layer

structure and air quality. The analyses are based on four year measurements, and long-term analysis of measurement

is quite limited in China. I would recommend it for publication after some improvements. Detailed comments are

listed below: 1. The authors should carefully check the language and grammar. For example, 'feather' is used many times, but it

should be 'feature'. Language problems are in other places as well. 2. Line 63-64: There is no evidence to support this point. Weak surface wind and stable boundary layer stratification

do not necessarily mean regional transport cannot happen. Many studies have confirmed the roles of regional

transport during haze. 3. The title of 3.2 'Evaluation against meteorological data' is not appropriate. Fig. 3 shows the characteristics of

meteorological variables for each classified type 4. Line 229-239: This explanation is not solid, at least not complete. It is more likely that the increasing stability

promotes the accumaltion of aerosols, and strong aerosol-radiaiton interactions inhibit photochemistry.

---

## Author Comment (AC1) · 21 Mar 2018

The comment was uploaded in the form of a supplement:
https://www.atmos-chem-phys-discuss.net/acp-2017-1046/acp-2017-1046-AC1-supplement.zip

---

## Author Comment (AC2) · 21 Mar 2018

The comment was uploaded in the form of a supplement:
https://www.atmos-chem-phys-discuss.net/acp-2017-1046/acp-2017-1046-AC2-supplement.zip

---

## Author Response (AR1)

**Response to Reviewers**

Dear Editors and Reviewers:

Thank you for your letter and the reviewers' comments regarding our manuscript entitled "Influence of boundary layer structure on air quality in Beijing: Long-term analysis based on self-organized maps" (ID: ACP-2017-1046). These comments have greatly improved the quality of our manuscript and are of important help to our future research. We have addressed the comments carefully and made changes accordingly which we hope satisfying the reviewers. The relevant changes are marked in red in the marked-up manuscript.

**Responses to the reviewer's comments:**

**Reviewer #1:**

This manuscript uses an unsupervised machine learning to understand the relationships between boundary layer structure and air quality. The analyses are based on four year measurements, and long-term analysis of measurement is quite limited in China. I would recommend it for publication after some improvements.

**Thank you for the positive comments. They encouraged us very much.**

Detailed comments are listed below:

1. The authors should carefully check the language and grammar. For example, 'feather' is used many times, but it should be 'feature'. Language problems are in other places as well.
**Reply:** Thanks for the comment. We checked the language and grammar and re-wrote some sentences in the revised manuscript.

2. Line 63-64: There is no evidence to support this point. Weak surface wind and stable boundary layer stratification do not necessarily mean regional transport cannot happen. Many studies have confirmed the roles of regional transport during haze.
**Reply:** Thanks for the useful comment. As you said, the transport can occur at any time in the presence of wind, regardless of wind speed. In our previous manuscript, the relative contributions of local accumulation and regional transport were discussed by comparing the ratio of $CO/SO_2$ under the different stability conditions. However, in the revised process, we found that the ratio of $CO/SO_2$ may be not an efficient indicator for long-term analysis because $SO_2$ can be reduced by chemical processes. The previous conclusions based on the ratio of $CO/SO_2$ were indefinite and therefore removed in the revised paper.

3. The title of 3.2 'Evaluation against meteorological data' is not appropriate. Fig. 3 shows the characteristics of meteorological variables for each classified type

**Reply: Thank you for the excellent comments.** We merged the meteorological analysis into one section and changed the title in the revised paper. Accordingly, we changed the other sections' title.

"3.1 Self-organized boundary layer meteorology" (Line 157).

"3.2 Implementing the SOM-based ABL classification scheme for urban air quality assessment" (Line 215).

"3.3 Quantifying the contribution of ABL anomaly to typical-month $PM_{2.5}$ air quality" (Line 304).

4. Line 229-239: This explanation is not solid, at least not complete. It is more likely that the increasing stability promotes the accumulation of aerosols, and strong aerosol-radiation interactions inhibit photochemistry.

**Reply: Thanks for the useful comment.** We explained the response of $O_3$ to different near-surface stability from both physical and chemical perspectives in the revised paper.

"However, increasing atmospheric stability has the opposite effect on near-surface $O_3$ concentrations. Since aerosols can absorb and reflect solar radiation and thereby inhibit the photochemical production of $O_3$ (Gao et al., 2016;Kaufman et al., 2002), the lowest $O_3$ concentration is observed in Node 9. In addition, considering that ozone is mainly produced in the upper ABL, near-surface $O_3$ should be strongly modulated by down-mixing processes (Tang et al., 2017b;Tang et al., 2017a). In this light, the varying daytime $O_3$ peaks across the ABL types can be partly attributed to the various magnitudes of vertical mixing. This is supported by daytime BLH. As have shown in Fig. 4, the daytime BLH is highest in Node 1, followed by Node 7, Node 3 and the lowest in Node 9. Such ordering is generally consistent with the daytime $O_3$ peaks in these types. Due to the persistent down-mixing caused by strong wind shears, the near-surface $O_3$ remains a relatively high nocturnal concentration (e.g. about 45 μg/m$^3$ in winter) in Node 1. In contrast, the stable nocturnal conditions (e.g., Nodes 9, 7 and 3) are commonly associated with low $O_3$ concentration (e.g. about 16 μg/m$^3$ in winter) due to the lack of vertical mixing, as well as the strong chemical titration by NO emitted from vehicles." (Lines 271-282)

**Reviewer #2:**

General comments

The paper deals with the influence of boundary layer structure on air quality using 4-year observations. The article presented very interesting study between air pollutants and meteorology, and the study built on very good meteorological measurements data. The paper is certainly worth of publishing as the study itself is extremely interesting. However, some improvements/corrections are suggested.

**Thank you very much for the positive comments. We have addressed each of the concerns you've brought up here through our responses below.**

Specific comments

1. The title should be modified. Throughout the manuscript, I haven't found the description concerning the calculation of the BLH. Therefore, the title using boundary layer structure is not exactly correct. Maybe atmospheric stability or boundary layer meteorology are much better. In addition, 4-year is not long-term for the observation period. In section 3.4, the authors also discussed the impact of the aerosols on BLH. However, the title just represented the influence of boundary layer meteorology on air quality. I suggest removing section 3.4 or please revise the title correspondingly.

**Reply: Thanks for the useful comments.** We changed the title in the revised paper. In addition, according to your suggestion, we removed section 3.4 in the revised paper.

"Self-organized classification of boundary layer meteorology and associated characteristics of air quality in Beijing" (Lines 1-2)

2. The radiosondes the authors used mainly represent the stable condition of the boundary layer. However, the air pollutants were observed in the whole day. Especially, the AOD data were just observed during the daytime. As well known, the boundary layer height will develop rapidly during the daytime. Lacking of the daytime observations of boundary layer structures, how about the creditability of the relationship of air pollutants and boundary layer meteorology?

**Reply: Thanks for the excellent comments.** We estimated the daytime boundary layer height (BLH) with parcel method to detect the boundary layer development after sunrise in the revised manuscript. In general, the daytime boundary layer heights are relatively flat after an extremely stable night, reflecting an insufficient space for vertical mixing in the day. These BLH results together with the classified ABL types jointly supported the analysis on the relationship of air pollutants and boundary layer meteorology.

"To detect the boundary layer development after sunrise, daytime boundary layer height (BLH) is estimated with parcel method (Holzworth, 1964, 1967), i.e., intersecting each day's 08:00 radiosonde potential temperature ($\theta$) profile at Beijing Observatory with each hour's (from 09:00 to 15:00) surface $\theta$ values, which are calculated from surface air temperature observations. As shown in Fig. 4, the BLH on the days following a strong stable night (i.e., Node 1) is relatively flat, reflecting an inadequate development of daytime boundary layer. Similarity, Node 3 is also followed by a flat daytime BLH variation. The maximum BLH in these two types are lower than 900 m, indicating a limited space for vertical mixing in the day. In contrast, the afternoon BLH in Node 7 can reach up to 1100 m; this mixing depth is conducive to dilute the pollutants accumulated in the previous night. In Node 1, the convective boundary layer develops well, and its maximum height on average exceeds 1500 m, far higher than the values in other types." (Lines 201-210)

[Figure]

Figure 4. Daytime boundary layer height (BLH) estimated for the four typical ABL types (i.e., Nodes 1, 3, 7 and 9).

3. The instruments of US Embassy for PM are not the same as the MEP. Do you consider the differences?

**Reply: Thanks for the comment.** The $PM_{2.5}$ values in the two different dataset showed high consistence. We clarified it in the revised paper.

"The mass concentrations of atmospheric pollutants (including $PM_{2.5}$, $O_3$, $NO_2$, $SO_2$ and CO) over Beijing during the period from 2013 to 2017 are obtained from the Ministry of Environmental Protection of the People's Republic of China (http://datacenter.mep.gov.cn/). In addition, hourly $PM_{2.5}$ measured at the Beijing US Embassy (http://www.stateair.net/) are also used in this study. The

PM$_{2.5}$ values in the two datasets show a well consistence with a mean correlation coefficient of 0.94. The mean hourly standard error of PM$_{2.5}$ across sites changes little from 12.6 to 12.9 after the inclusion of US Embassy." (Lines 112-117)

4. For the boundary layer meteorology, potential virtual temperature and Richardson number are much better indicators to do the SOM analysis. Why the authors using temperature to do the SOM analysis? I suggest using potential virtual temperature to do the classification of different nodes.

**Reply: This excellent comment is highly appreciated.** We used virtual potential temperature to perform the classification of boundary layer meteorology in the revised manuscript.

"We construct a 3 $\times$3 SOM matrix for daily virtual potential temperature deviation profiles, and the self-organized output shown in Fig. 1 represents nine ABL types (i.e., SOM nodes)." (Lines 159-160)

[Figure]

Figure 1. The 3 $\times$ 3 SOM output for radiosonde-based virtual potential temperature ($\theta_v$) deviation profiles observed at the Beijing Observatory. SOM nodes are shown in red, with the corresponding individual profiles in grey. For reference, the overall average $\theta_v$ deviation profile and 25th and 75th percentile profiles are shown in cyan. The top-right shows the occurrence cases and frequency of each SOM node.

5. How to evaluate atmospheric stability? Do you have the quantitative basis? Node 3 represents slow wind, high humidity and more stable than node 7.

**Reply: Thanks for the useful comment.** We used the virtual potential temperature gradient profile to quantify the atmospheric stability at different heights (the larger gradient suggests the stronger stability) in the revised paper. We discussed the atmospheric stability throughout the ABL. For Node 3, it represents a strongest stability in the upper ABL compared to other nodes.

"Figure 3 displays the average profiles of wind speed, relative humidity, virtual potential temperature gradient according to the ABL types…" (Lines 181-194)

[Figure]

Figure 3. Profiles of average wind speed (WS), relative humidity (RH) and virtual potential temperature gradient ($\Delta\theta_v/\Delta z$) corresponding to individual ABL types (i.e., SOM nodes) at the Beijing Observatory. The black, green and red labels of the horizontal axis correspond to $\Delta\theta_v/\Delta z$, WS and RH, respectively.

6. Because emissions are different in different seasons, I suggest discussing the concentrations of air pollutants in different nodes for each season.

**Reply: Thanks for the useful comments.** We discussed the concentrations of air pollutants in different nodes separately by seasons, and therefore re-wrote the results in the section 3.2 of the revised paper. (Lines 215-302)

In addition, to increase the sample size, we included the observations in 2017 into analysis in the revised paper. (Lines 94-95 and Lines 112-115)

7. For the boundary layer ozone analysis, please refer Tang et al., 2017a, 2017b.

**Reply: Thanks for the comments.** We read the two papers carefully and referred them in our boundary layer ozone analysis.

"In addition, considering that ozone is mainly produced in the upper ABL, near-surface $O_3$ should be strongly modulated by down-mixing processes (Tang et al., 2017b;Tang et al., 2017a). In this light, the varying daytime $O_3$ peaks across the ABL types can be partly attributed to the various magnitudes of vertical mixing."(Lines 273-276)

Technical comments

1. Line 48. Beijing has two directions adjacent to mountains. The west one is Tai hang Mountains and two north one is Yan Mountains.

**Reply:** We corrected it in the revised manuscript.

"Beijing, the capital of China, is geographically located at the northwestern border of the Great North China Plain. This city is surrounded by the Yan Mountains to the north and the Taihang Mountains to the west, with the Bohai Sea to the 160 km southeast (Fig. 1)."(Lines 48-50)

2. Line 56. Ground-based remote sensing.

**Reply:** We corrected it in the revised paper.

"In addition, numerous intensive ABL measures were conducted using other approaches such as mooring boats, airplane, and ground-based remote sensing (Tang et al., 2015;Zhu et al., 2016;Zhang et al., 2009;Hua et al., 2016)." (Lines 60-62)

3. The second paragraph in the introduction section is too long. Please separate it.

**Reply:** We separated the second paragraph in the revised paper. (Lines 48-68)

4. Section 3.3 is too long, please use subtitle to separate it into several small parts.

**Reply: Thanks for the comments.** We shorten this section in the revise process and therefore did not use subtitle in the revised paper. In this section, we removed the discussion about local contribution derived from ratio of $CO/SO_2$. Considering $SO_2$ are chemical reactive, the chemical process can disrupt the analysis of relative contribution of local accumulation and regional transport.

Other changes:

1、To reflect more daytime concentration after the different stability nights, the daily concentration is performed afternoon-to-afternoon (15:00 h-15:00 h) in the revised paper. In the previous manuscript, the daily concentration was calculated from noon-to-noon (12:00 h-12:00 h).

2、We included the observations in 2017 into analysis in the revised paper. Some according change occurred in the section 3.3 (Quantifying the contribution of ABL anomaly to typical-month $PM_{2.5}$ air quality). Particularly, the results from the meteorology-to-environment method had some difference after the inclusion of observations in 2017. Given that, we updated the results in the revised manuscript.

3、We excluded $PM_{10}$ in the revised paper because to a large extent its characteristics can be represented by $PM_{2.5}$.

**Self-organized classification of boundary layer meteorology and associated characteristics of air quality in Beijing**

Zhiheng Liao[a], Jiaren Sun[a, b]*, Jialin Yao[c], Li Liu[a], Haowen Li[a], Jian Liu[a], Jielan Xie[a], Dui Wu[d], Shaojia Fan[a]*

[a] School of Atmospheric Sciences, Sun Yat-sen University, Guangzhou, Guangdong, China;

[b] South China Institute of Environmental Sciences, Ministry of Environmental Protection of the People's Republic of China, Guangzhou, Guangdong, China;

[c] Weather Modification Office of Shanxi Province, Taiyuan, Shanxi, China;

[d] Institute of Mass Spectrometer and Atmospheric Environment, Jinan University, Guangzhou, Guangdong, China.

* Address correspondence to S. Fan or J. Sun, School of Atmospheric Sciences, Sun Yat-sen University, Guangzhou, Guangdong, China. Telephone: +86 020 8411 5522.

E-mail: eesfsj@mail.sysu.edu.cn (S. Fan); sunjiaren@scies.org (J. Sun).

**Abstract**

Self-organizing maps (SOMs; a feature-extracting technique based on an unsupervised machine learning algorithm) are used to classify  atmospheric boundary layer (ABL) meteorology  over Beijing  through detecting topological relationships among the 5-year (2013–2017) radiosonde-based virtual potential temperature  profiles. The  classified ABL types are then examined in relation to near-surface  pollutant  concentrations  to understand the  modulation effects of  ABL  structures on Beijing's air quality. Nine ABL types (i.e., SOM nodes) are obtained through SOM classification technique, and each  is characterized  with distinct dynamic and thermodynamic conditions. In general, the self-organized ABL types are able to distinguish between high and low loadings of near-surface pollutants. The average concentrations of $PM_{2.5}$, $NO_2$ and CO dramatically increased from the near neutral (i.e., Node 1) to strong stable conditions (i.e., Node 9) during all seasons except summer. Since extremely strong stability can isolate the near-surface observations from the influence of elevated $SO_2$ pollution layers, the highest average $SO_2$ concentrations are typically observed in Node 3 (a layer with strong stability in the upper ABL) rather than Node 9.

 In contrast, near-surface $O_3$ shows an opposite dependence on atmospheric stability, with the lowest average concentration in Node 9. ~~The ABL controls on diurnal cycles of pollutants are as follows: (1) elevated inversion enhances the afternoon baseline; and (2) surface inversion improves the evening increment. Comparing the $CO/SO_2$ ratios for the different ABL types demonstrates that the local contribution increases with enhanced static stability near the ground, and it is the stable ABL stratification rather than weak surface wind that confines the regional contribution. Due to regional transport, node 3 (dominated by elevated inversion with high relative humidity) corresponds to the most severe columnar aerosol pollution, characterized by the highest optical depth (1.22) and volume concentration (0.30 $\mu m^3/\mu m^2$). The larger aerosol radiative forcing (ARF) within the atmosphere (> 60 $W/m^2$) in nodes 3, 6 and 9 is likely to strengthen the atmospheric stability and thus induce a positive feedback loop for causing high surface pollution.period~~ months (i.e., January 2013, December 2015 and December 2016) suggests that the ABL types are the primary drivers of day-to-day variations in Beijing's air quality. Assuming a fixed relationship between ABL type and $PM_{2.5}$ loading for different years, the relative (absolute) contributions of the ABL anomaly to elevated $PM_{2.5}$ levels are estimated to be 58.3 % (44.4 $\mu g/m^3$) in January 2013, 46.4 % (22.2 $\mu g/m^3$) in December 2015, and 73.3 % (34.6 $\mu g/m^3$) in December 2016.

**1 Introduction**

The atmospheric boundary layer (ABL) is the section of atmosphere that responds directly to the flows of mass, energy and momentum from the earth's surface (Stull, 1988). Since most air pollutants are emitted or chemically produced within this layer, its evolution plays an important role in transport dispersion and deposition of air pollutants (Chen et al., 2012;Fan et al., 2008;Whiteman et al., 2014;Platis et al., 2016;Wolf et al., 2014;Wu et al., 2013). The ABL structure is determined by complex interactions between atmosphere static stability and those mechanical processes (such as wind shear from synoptic or terrain-induced flows) (Stull, 1988;Chambers et al., 2015b). These processes can operate at a variety of different heights and temporal scales, and their dominance may vary considerably with height and time at any given location (Salmond and McKendry, 2005). This makes it very difficult to observe and predict the transport and diffusion of air pollutants within the ABL (Chambers et al., 2015b;Chambers et al., 2015a), particularly in those complex-terrain regions such as Beijing.(Stull, 1988)Therefore, characterizing typical ABL conditions associated with high pollution levels helps to better understand the role of ABL in governing the transport and distribution of pollutants in the atmosphere.

Beijing, the capital of China, is suffering serious air pollution problems. This city is geographically geographically located at the northwestern border of the Great North China Plain. This city and has is surrounded by the Yan Mountains to the north and the Taihang Mountains to the westthree directions that are adjacent to mountains., with the Bohai Sea to the 160 km southeast (Fig. 1). Under favorable weather conditions (e.g., stagnant weather), The closest coast from the city of Beijing is the Bohai Sea, which is 160 km southeast of the city. Tterrain-related circulations can therefore be well developed over Beijing and its surroundings under favorable weather conditions, leading to a complex ABL ABL thermodynamic structure, which is thought to substantially affect Beijing's Beijing's air quality (Hu et al., 2014;Miao et al., 2017;Ye et al., 2016;Gao et al., 2016;Xu et al., 2016). With high emissions of air pollutants from anthropogenic sources, Beijing is suffering serious air pollution problems and the pollution can be even more severe when southwesterly and southeasterly winds prevail within the ABL (Chen et al., 2008;Ye et al., 2016;Zhang et al., 2014;Zhang et al., 2012a).

Several studies used tower-based observations to investigated the interactions between boundary layerABL meteorologydynamics and air pollution formationquality in Beijing using tower-based observations –(Sun et al., 2013;Sun et al., 2015;Guinot et al., 2006;Guo et al., 2014). However, the results are not ideal because the tower-based observationsthey have a low observational height (325 m). In addition, nNumerous intensive ABL measures were conducted using other approaches, such as mooring boats, airplane, and ground-based remote sensing (Tang et al., 2015;Zhu et al., 2016;Zhang et al., 2009;Hua et al., 2016). However, sSince these approaches are complex, expensive and labor intensive, they are often restricted to the duration of specific research campaigns and hence their results may be considered 'unrepresentative'. Overall, the existing knowledge of linkages between ABL meteorologystructure and air quality in Beijing is drawn largely from either low observational height or short observational duration. Due to the lack of long-term effective observations; therefore, the common patterns of the influence influence of the changing ABL ABL structures on Beijing's air quality remains relatively unclearunclear and need to be further studied (Quan et al., 2013;Miao et al., 2017;Guo et al., 2014). For example, many case studies (Jia et al., 2008;Zheng et al., 2015;Hua et al., 2016;Li et al., 2016) claimed that rapid growth of PM$_{2.5}$ in Beijing is mainly attributable to the regional transport of the polluted air mass. This view is occasionally questionable, as it is known that the polluted episodes tend to occur with a weak surface wind and stable boundary

Meanwhile, the routine radiosondes are not being fully utilized to investigate urban pollution issues. The advantage of radiosondes over the other approaches seems to be their length, which usually spans several decades. For a long time, it was challenging to reduce the wealth of radiosonde data to characterize the ABL structure, and therefore, radiosondes remain a very limited use in case studies (Ji et al., 2012;Zhao et al., 2013;Gao et al., 2016). Recently, self-organizing maps (SOMs; a feature-extracting technique based on an unsupervised machine learning algorithm) (Kohonen, 2001) were introduced to investigate the ABL thermodynamic structure, indicating the capabilities of SOMs in  feature extraction from a large dataset of the ABL measurements (Katurji et al., 2015). In fact, the SOM has  an increasing in atmospheric and environmental sciences during the past several years (Jensen et al., 2012;Jiang et al., 2017;Gibson et al., 2016;Pearce et al., 2014;Stauffer et al., 2016), including a radiosonde-based application  in South Africa (Dyson, 2015). However, there is thus far no SOM application in air pollution-related ABL structure research. It is expected  such a new analytical approach can tap the potential of routine radiosondes to  reveal the ABL mechanism of air pollution in Beijing.

This study investigates the influence of ABL  meteorology on Beijing's air quality  based on the SOM application  to 5 years (20132017) of routine meteorological radiosondes. First, we use the SOM technique to classify the state of ABL through detecting topological relationships among the radiosonde-based virtual potential temperature profiles (see section 3.1). Then, we, we provide a visual insight into near-surface pollutant variations  under various ABL  types and discuss the potential physical mechanisms behind their relationships (see section 3.2–3.3). It is expected that such an association between air quality and ABL type could provide local policy makers with useful information for improving the predictions of urban air quality.

**2 Materials and methods**

**2.1 Data preparation and preprocessing**

The recent 5-year (2013–2017) radiosonde data observed at the Beijing Observatory (39.81$°$N, 116.48$°$E, WMO station number 54511) were collected from the University of Wyoming (http://weather.uwyo.edu/).  The radiosondes were  launched twice a day (08:00 and 20:00  Local Time , corresponding to the morning and evening, respectively) and  provided atmospheric sounding data (profiles of temperature, relative humidity, wind speed, etc.) at the mandatory pressure levels (e.g., surface, 1000, 925, 850, 700 hPa) and additional significant levels. In addition, the hourly near-surface meteorological parameters (including temperature, wind speed and relative humidity, etc.) in 2013–2016 were collected from the Beijing Meteorological Bureau.

We chose the 2000 m above ground level  as the upper limit of the ABL based on a number studies investigating the ABL height over Beijing or North China (Tang et al., 2016;Guo et al., 2016;Miao et al., 2017). This height  exceeded the top of the ABL  in most cases, and therefore, most ABL processes influencing the near-surface air quality  were included in the analysis herein. In our study period, the average number of data points in radiosonde profiles was 3.7 below 500 m and 10.1 below 2000 m. Despite the coarse resolution, the profile shapes were enough for SOM technique to classify the state of ABL. Previous radiosonde-based study indicated that surface temperature inversions occur frequently in the eastern China (Li et al., 2012), suggesting that all of the two-time radiosondes mainly represent the nocturnal stable ABL.  To keep a whole night, the daily ABL profiles  were composited from the radiosondes at 20:00 and 08:00 of the next day.

The mass concentrations of atmospheric pollutants (including  PM$_{2.5}$, O$_3$, NO$_2$, SO$_2$ and CO) over Beijing during the period from 2013 to 2017 are obtained from the Ministry of Environmental Protection of the People's Republic of China (http://datacenter.mep.gov.cn/). In addition, hourly PM$_{2.5}$ measured at the Beijing US Embassy (http://www.stateair.net/) are also used in this study. The PM$_{2.5}$ values in the two datasets show a well consistence with a mean correlation coefficient of 0.94. The mean hourly standard error of PM$_{2.5}$ across sites changes little from 12.6 to 12.9 after the inclusion of US Embassy. Hourly concentrations are calculated for the Beijing urban area by averaging concentrations from nine urban sites (including Dongsi, Guanyuan, Tiantan, Wanshouxigong, Aotizhongxin, Nongzhanguan, Gucheng, Haidianwanliu and US Embassy). To maintain consistency with ABL classification, the daily pollutant concentration is then performed  afternoon-to-afternoon (15:00 h–15:00 h), in order to include one whole night in each 24-h period.

~~In addition to near-surface observations, columnar aerosol parameters (including aerosol optical depth (AOD), Ångström exponent (AE), single scattering albedo (SSA), volume particle size distribution (d*V*/dln*R*), aerosol radiative forcing (ARF) and so on) are also collected from the AERONET Beijing (39.98°N, 116.38°E) and Beijing CAMS (39.93°N, 116.32°E) sites. The level 2.0 quality-assured columnar aerosol data from 2013 to 2016 are downloaded from the AERONET data archive (http://aeronet.gsfc.nana.gov). The size distribution is retrieved in 22 logarithmically equidistant bins in a range of sizes from 0.05 to 15 μm through a combined spherical and spheroid particle model (Dubovik and King, 2000;Dubovik et al., 2006).~~

**2.2 Self-organizing maps technique**

The SOM is thought to be an ideal tool for feature extraction because the input data are treated as a continuum without relying on correlation, cluster or eigenfunction analysis (Liu et al., 2006). Since Kohonen (1982) first proposed SOM, it has been widely used for data downscaling and visualization in various disciplines (Jensen et al., 2012;Katurji et al., 2015;Dyson, 2015;Stauffer et al., 2016;Pearce et al., 2014;Jiang et al., 2017). In this study, the SOM is introduced to classify the ABL types through detecting topological relationships among the 5-year (2013–2017) radiosonde-based virtual potential temperature profiles. Since the SOM is sensitive to the virtual potential temperature value, the deviation profiles, which are determined by subtracting the mean virtual potential temperature of each profile from each level, are used as the SOM input.

The training of SOM is an unsupervised, iterative procedure, and the result is a matrix of nodes (i.e., types) that represent the input data. The following provides a simple introduction about the SOM algorithm, and the details can be found in Kohonen (2001).  To learn from the input data, every SOM node has a parametric reference vector with which it is associated, and these reference vectors are randomly generated. After initialization of the reference vectors, a stochastic input vector is compared to every reference vector, and the closest match, named the best-matching unit, is determined by the smallest Euclidean distance. Each reference vector is then updated so that the best-matching unit and its neighbors become more like the input vector. Whether or not a reference vector learns from the input vector is determined by the neighborhood function. Only reference vectors that are topologically close enough to the best-matching unit will be updated according to the SOM learning algorithm.

 The first step of SOM training is to determine a matrix size of nodes for initializing the reference vectors. This step is performed subjectively and depends on the degree of generation required (Lennard and Hegerl, 2015). We test several SOM matrixes, and finally select a $3 \times 3$ matrix, because it captured unique profiles without the profiles being too general as with a smaller matrix, or being too similar as with a larger matrix.  In addition, the batch mode is chosen to execute the SOM algorithm, because it is much more computationally efficient compared to the sequence mode. The other user-defined settings in the SOM software are set at the default, such as the hexagon topology, Gaussian neighborhood function, etc. The SOM code used in this study is sourced from the MATLAB SOM Toolbox, which is freely available from http://www.cis.hut.fi/projects/somtoolbox/.

**2.3 Measuring the discriminative power of SOM technique for pollution assessment**

The Kruskal-Wallis one-way analysis of variance is used as a non-parametric method to test the difference of pollutant concentrations among the various ABL types. A 1% significance level is used and hereafter denoted as *KW* in Sect. 3.2. Furthermore, the coefficients of variation (*CV*) of pollutant means across the various ABL types are also used to examine the discriminative power of the SOM technique for pollution assessment.

**3 Results and discussion**

**3.1 Self-organized boundary layer meteorology**

We construct a $3 \times 3$ SOM matrix for daily virtual potential temperature deviation profiles, and the  self-organized output shown in Fig. 1 represents nine ABL types (i.e., SOM nodes).

On the SOM plane, the most notable feature is adjacency of like types (e.g.,  Nodes

1 and 2) and the separation of contrasting types (e.g.,  Nodes 1 and 9).

Such ordering is a feather of the SOM algorithm (i.e., 'self-organized') , which  allows us to distinguish the unique characteristics of nodes through the variation of specific features across the SOM plane. According to the ordering feature, the

SOM nodes in the four corners (i.e., Nodes 1, 3, 7 and 9) can be thought of as the typical types and the others can be considered as transitional types. The four typical ABL types have a relatively higher occurrence frequency (>

10%), with the highest frequency associated to Node 1 (23%). Furthermore, the seasonal statistical results (Fig. 2)

reveal that these self-organized ABL types exhibit a strong seasonality. For example, Node 3 occurs more frequently in winter and autumn, while Node 1 has a relatively higher occurrence in spring and summer.

[Figure]

**Figure 1. The 3 ×3 SOM output for radiosonde-based  temperature ($\theta_v$) deviation profiles**

**observed at the Beijing Observatory. SOM nodes are shown in red, with the corresponding individual**

**profiles in grey. For reference, the overall average $\theta_v$ deviation profile and 25th and**

**75th percentile profiles are shown in cyan. The top-right shows the occurrence cases and frequency of each SOM node.**

[Figure]

**Figure 2.** **Relative frequency of individual ABL types (i.e., SOM nodes) in all four seasons. Winter (DJF); Spring (MAM); Summer (JJA); Autumn (SON).**

~~The SOM classification reveals that for the whole study period, the ABL is dominated by near neutral to strong stable conditions, as none of the SOM nodes fall within the unstable category (i.e., super-adiabatic condition). The results are reasonable, considering the daily temperature profile is composited from 20:00 and 08:00 measurements. According to the SOM ordering feather, the SOM nodes in four corners (i.e., nodes 1, 3, 7 and 9) can be thought of as the typical ABL types and the others can be considered transitional ABL types. It is clear from the individual profiles in Fig. 1 that node 1 represents the well-mixed (near-neutral) condition with no temperature inversion, node 3 indicates the ABL type dominated by elevated inversion, node 7 indicates the ABL type dominated by surface inversion, and node 9 represents the ABL type associated with multiple inversions (i.e., including surface and elevated inversions).~~

~~Frequency analysis of the nine ABL types indicates that the frequency distribution across the types is quite varied from the expected 11.1 %, with the occurrence frequency showing a 5:1 range from the most frequent type (node 1) to the least frequent type (node 5). The higher frequency types are presented on the outer portions of the SOM plane, while lesser frequency types are presented closer towards the center (top-right in Fig. 1). The most dominant types are nodes 1 and 3, and their occurrence frequencies reach 22 % and 20 %, respectively. As~~

synoptic circulations change with the seasons over Beijing, the ABL types are expected to correspond to seasonality. The number of profiles from each season in each ABL type is expressed as a percentage and is shown in Fig. 2. All of the types exhibit strong seasonality. For example, node 1 has the highest occurrence in spring (29.4 %) and the lowest occurrence in autumn (13.7 %); node 9 presents the highest occurrence in winter (16.3 %)

and the lowest occurrence in summer (4.9 %).

3.2 Evaluation against meteorological data

Figure 3 displays the average profiles of wind speed, relative humidity, virtual potential temperature gradient according to the ABL typesFig. 3 shows the average vertical profiles of potential temperature, wind speed and relative humidity corresponding to each ABL type. As seen in Fig. Clearly3, each of the the self-organized ABL

types is associated withfeatures distinct dynamic and thermodynamic conditions within the ABL, suggesting the

SOM technique is feasible to classify the boundary layer meteorologys. Since the classification is based on the twice-daily radiosondes, the resulting ABL types are dominated by near neutral to strong stable conditions, and none of the types fall within the unstable category (i.e., $\Delta\theta_v/\Delta z$ <0). While Node 3 features the strong static stability in the upper ABL (the large $\Delta\theta_v/\Delta z$ values), Node 1 represents a near neutral ABL condition with the lowest $\Delta\theta_v/\Delta z$ values and the highest wind shears in the lower ABL. In contrast, Node 7 corresponds to a moderate static stability in the lower ABL, and Node 9 relates to a strong static stability. Particularly, the virtual potential temperature gradient in Node 9 remains a high level (> 0.7$^o$C/100m) from surface to approximately 800

m, indicating a strong and deep surface temperature inversion developed in this type. In addition, due to the strong surface inversion, vertical mixing is suppressed, resulting in a strong decreasing gradient in humidity profiles.

Overall, the SOM classification scheme reveals a significant coupling between dynamic and thermal effects in the

ABL, which is expected to considerably impact the near-surface air quality. The potential temperature profiles vary from near neutral conditions to strong stable conditions, and this change is closely related to the variance in wind speed, suggesting a strong coupling between the dynamic and thermal effects. The two extreme types (nodes

1 and 9) provide a very useful example. Node 9 is a very strong stable profile, and the wind speeds are very low in the lower ABL. In contrast, node 1 is a well-mixed (near neutral) profile and it corresponds to significantly higher wind speeds throughout the ABL. In addition, when the stability of the atmosphere is strong, vertical mixing is suppressed and winds in the lower ABL become decoupled from the generally stronger wind aloft. This allows moisture, fogs, low clouds and other scalars to build up within the stable layer. As a result, the stable ABL types usually correspond to high RH in the lower ABL.

[Figure]

**Figure 3. Profiles of average  wind speed (WS) , relative humidity (RH) and**

**virtual potential temperature gradient ($\Delta\theta_v/\Delta z$) corresponding to individual ABL types (i.e., SOM**

**nodes) at the Beijing Observatory. The black, green and red labels of the**

**horizontal axis correspond to $\Delta\theta_v/\Delta z$ , WS and RH, respectively.**

To detect the boundary layer development after sunrise, daytime boundary layer height (BLH) is estimated with parcel method (Holzworth, 1964, 1967), i.e., intersecting each day's 08:00 radiosonde potential temperature ($\theta$) profile at Beijing Observatory with each hour's (from 09:00 to 15:00) surface $\theta$ values, which are calculated from surface air temperature observations. As shown in Fig. 4, the BLH on the days following a strong stable night (i.e., Node 1) is relatively flat, reflecting an inadequate development of daytime boundary layer. Similarity, Node 3 is also followed by a flat daytime BLH variation. The maximum BLH in these two types are lower than 900 m, indicating a limited space for vertical mixing in the day. In contrast, the afternoon BLH in Node 7 can reach up to 1100 m; this mixing depth is conducive to dilute the pollutants accumulated in the previous night. In Node 1, the convective boundary layer develops well, and its maximum height on average exceeds 1500 m, far higher than the values in other types.

[Figure]

**Figure 4. Daytime boundary layer height (BLH) estimated for the four typical ABL types (i.e., Nodes 1, 3, 7 and 9).**

**3.2 Implementing the SOM-based ABL classification scheme for urban air quality assessment**

In the previous section, it was seen that the SOM classification scheme is an effective tool for delineation between various dynamic and thermodynamic structures within the ABL. As a further evaluation, we implement the new classification scheme to quantify changes in various urban pollutant concentrations as a function of ABL types. Since the pollutant emissions have a strong seasonality over Beijing and its surroundings, the analyses are performed for winter (December to February), spring (March to May), summer (June to August), and autumn (September to November), respectively. Figure 5 shows the statistical distributions of daily pollutant concentrations according to the nine ABL types, along with the results of Kruskal-Wallis test and the coefficients of variation of pollutant means across the various types. Figure 6 displays the type-average pollutant diurnal patterns composited for the four typical ABL types (i.e., Nodes 1, 3, 7 and 9).

~~The concentrations of gaseous and particulate pollutants in the atmosphere are governed by the rate at which they are emitted from their respective sources, lost by various sink mechanisms, and characteristics of the atmospheric volume into which they mix. While the mixing volume is determined primarily by the boundary layer structure, the chemical transformation also depends on boundary layer meteorology in some cases. In the previous section, it was seen that the SOM technique is an effective tool for classifying boundary layer structures. In this section, we used the classification technique to quantify the influence of the boundary layer structure on near-surface air quality.~~

 The Kruskal-Wallis test demonstrates that the self-organized ABL types are able to distinguish between high and low loadings of air pollutants, with $KW$ < 1% in all seasons (except for summertime $SO_2$ with a $KW$ value of 1.5%). Furthermore, it is found that the SOM technique has a stronger discriminative power for $SO_2$, $PM_{2.5}$ and CO assessments, which is supported by relatively higher $CV$ values ($CV$>0.30). According to the seasonal $CV$ values, this discriminative power shows a following seasonal ordering: winter > autumn > spring > summer. Particularly, the wintertime $CV$ value in $PM_{2.5}$ assessment reaches the maxima (0.56), indicating an extremely strong dependence of $PM_{2.5}$ air quality on the changing ABL meteorology in winter. In summer, the stable nocturnal ABL develops later due to the longer day (Li et al., 2012), and hence avoids the larger daytime pollutant emissions, particularly the traffic peak emissions. In the absence of larger sources, the nocturnal stable layers exert a limited influence on near-surface air quality; therefore, the classified ABL types have relatively weakened discriminative power for summertime pollution assessments. In addition, wet depositions (more precipitation in summer) play an important role in modulating summertime air quality, and to some degree disrupt the linkages between ABL type and air quality.

[Figure]

**Figure 5. Daily concentrations of (a) $PM_{2.5}$, (b) $O_3$, (c) $NO_2$, (d) $SO_2$, and (e) CO in Beijing for all nine ABL**

**types separately in (1) winter, (2) spring, (3) summer, and (4) autumn. The solid dots denote the mean. The**

**box and whisker plot presents the median, the first and third quartiles, and the 5th and 95th percentiles,**

**respectively. The upper numbers denote the results of Kruskal-Wallis test (*KW*) and the coefficients of**

**variation (*CV*) of pollutant means across the various types.**

In the case of $PM_{2.5}$, $NO_2$ and CO, the most stable atmospheric conditions (i.e., Node 9) are associated with dramatically increased near-surface pollutant concentrations in all seasons except summer. The wintertime average concentrations of $PM_{2.5}$, $NO_2$ and CO in Node 9 reach up to 197.2 μg/m³, 100.2 μg/m³ and 3.6 mg/m³, respectively. These values are 3–8 times higher than that in Node 1 (i.e., near neutral condition), with the highest increasing amplitude (a factor of 7.3) related to $PM_{2.5}$. As have known, Node 9 corresponds to the strongest nighttime stability in the lower ABL and the lowest daytime BLH. All of these ABL characteristics are extremely conduceive to the accumulation of air pollutants emitted near the ground. For Node 3, the concentrations of $PM_{2.5}$, $NO_2$ and CO are the second-highest compared to those of the other types. This ABL type features the strongest stability in the upper ABL, suggesting that processes operating at the different heights throughout the ABL may have a significant impact on near-surface pollutant concentrations.

The diurnal cycles of $PM_{2.5}$, $NO_2$ and CO are extremely pronounced under the strong stable conditions, although very reduced on the days with near neutral night. On average, the wintertime diurnal range of $PM_{2.5}$ increases from 18.2 $\mu g/m^3$ in Node 1 to 95.4 $\mu g/m^3$ in Node 9. The corresponding diurnal range increase for $NO_2$ is 18.9 to 33.6 $\mu g/m^3$, and for CO 0.2 to 1.7 $mg/m^3$. In Node 1, the diurnal variations are characterized by a weak two-peak pattern following the traffic rush hours, suggesting that traffic is the primary driver of these pollutants' diurnal cycles in Beijing (Liu et al., 2012). However, the diurnal effects of traffic emissions are significantly amplified by the ABL dynamics. It is clear that the more stable conditions near the ground, the higher peak concentrations are observed. In winter, the stable ABL conditions exert a more important influence on the evening traffic emissions, resulting in a broad evening peak. In contrast, the morning peak signature is much lower since the morning emission is counteracted by the destabilization of the ABL. However, as human activities begin earlier during the warm season, maximum concentrations in spring and summer are typically observed during the morning rush hours.(Pernigotti et al., 2007;Chambers et al., 2015b;Crawford et al., 2016;Chambers et al., 2015a)

However, increasing atmospheric stability has the opposite effect on near-surface $O_3$ concentrations. Since aerosols can absorb and reflect solar radiation and thereby inhibit the photochemical production of $O_3$ (Gao et al., 2016;Kaufman et al., 2002), the lowest average $O_3$ concentration is observed in Node 9. In addition, considering that ozone is mainly produced in the upper ABL, near-surface $O_3$ should be strongly modulated by down-mixing processes (Tang et al., 2017b;Tang et al., 2017a). In this light, the varying daytime $O_3$ peaks across the ABL types can be partly attributed to the various magnitudes of vertical mixing. This is supported by daytime BLH. As have shown in Fig. 4, the daytime BLH is highest in Node 1, followed by Node 7, Node 3 and the lowest in Node 9. Such ordering is generally consistent with the daytime $O_3$ peaks in these types. Due to the persistent down-mixing caused by strong wind shears, the near-surface $O_3$ remains a relatively high nocturnal concentration (e.g. about 45 $\mu g/m^3$ in winter) in Node 1. In contrast, the stable nocturnal conditions (e.g., Nodes 9, 7 and 3) are commonly associated with low $O_3$ concentration (e.g. about 16 $\mu g/m^3$ in winter) due to the lack of vertical mixing, as well as the strong chemical titration by NO emitted from vehicles.

As expected, the most stable conditions are associated with a dramatic increase in the mass concentrations of air pollutants (except $O_3$). On average, $SO_2$, $NO_2$, CO, $PM_{10}$ and $PM_{2.5}$ increase by 15.7 µg/m$^3$ (142 %), 44.3 µg/m$^3$ (119 %), 1.5 mg/m$^3$ (202 %), 91.6 µg/m$^3$ (119 %) and 95.9 µg/m$^3$ (218 %) from the near neutral ABL condition (i.e., node 1) to strong stable condition (i.e., node 9), respectively. The highest increasing amplitude is related to $PM_{2.5}$, suggesting fine particulate matters are likely accumulated from not only primary emissions but also secondary formation (Zhang and Cao, 2015). As we have shown, the more stable ABL conditions tend to correspond to high relative humidity in the lower ABL (Figs. 3 and 4). Additional enhancement in $PM_{2.5}$ can be expected under the humid condition, as it is known that the humidity related physicochemical formation of particles (such as hygroscopic growth, liquid-phase and heterogeneous reactions) can be intensified by high humidity values (Cheng et al., 2015;Cheng et al., 2016;Zheng et al., 2015).

Interestingly, increasing atmospheric stability has an opposite effect on near-surface $O_3$ concentrations. Since $O_3$ is produced by photochemical interactions between $NO_x$ (NO + $NO_2$) and volatile organic compounds (VOCs) (Seinfeld and Pandis, 2006), the boundary layer structure alters the $O_3$ level through modulation of its precursors ($NO_x$ and VOCs). The low $O_3$ level in the stable ABL can be explained by the strong titration reaction. Since $O_3$ is highly reactive, when trapped in a stable layer, surface titration by the NO emitted from vehicles can cause a rapid reduction in $O_3$ concentration. In previous studies, persistent low $O_3$ concentration were observed in the stable boundary layer condition in Beijing (Zhao et al., 2013). Conversely, when near-surface wind speeds are higher (near neutral condition such as node 1), $O_3$ is mixed downward from the overlying air mass, resulting in higher concentrations. Nevertheless, it is worth noting that the extremely high $O_3$ values (not shown) were also detected on very stable days (i.e., node 9), suggesting the complexity of $O_3$ behavior in response to the boundary layer structure (Tong et al., 2011;Haman et al., 2014).

To obtain a more in-depth understanding of the physical mechanisms behind the relationship between air quality and ABL structure, diurnal composite hourly concentrations of atmospheric pollutants are formed for each ABL type.

[Figure]

**Figure 6. Mean hourly composites of (a) PM2.5, (b) O3, (c) NO2, (d) SO2, and (e) CO in Beijing for the four typical ABL types separately in (1) winter, (2) spring, (3) summer, and (4) autumn**

The SOM based ABL classification scheme provides a consistent, gradual distinction in the diurnal cycles of surface air pollutants from near neutral to strong stable conditions. The composite diurnal evolutions of air pollutants in the four typical ABL types (i.e., nodes 1, 3, 7 and 9) are illustrated in Fig. 6. (Tang et al., 2017b;Tang et al., 2017a)

The highest average $SO_2$ concentrations are typically observed in Node 3, but occasionally in Node 9. Over the North China Plain, high stacks are emitting a significantly larger amount of $SO_2$ compared to small stacks (Zhao et al., 2012). The frequent surface temperature inversions together with the large $SO_2$ emissions from higher stacks favor the formation of elevated $SO_2$ pollution layers over Beijing (Chen et al., 2009). If sufficiently strong, the surface temperature inversion can even isolate near-surface observations from the influence of elevated pollution layers (Salmond and McKendry, 2005). This explains the commonly lower near-surface $SO_2$ concentration in

Node 9 than that in Node 3. However, after a stable night, the burst of turbulent activity in the morning coincides with a rapid increase in near-surface $SO_2$ concentration, resulting in a pre-noon peak. Since there is no significant increase in $SO_2$ emission at the surface at that time, the result strongly suggests that the $SO_2$ peaks are due to the downward mixing from the elevated $SO_2$ pollution layers. Regarding the physical mechanism of the noontime-peak $SO_2$ pattern, Xu et al. (2014) have made a detail explanation in a previous study. Nevertheless, the wintertime $SO_2$ concentration signature does not always show a distinct pre-noon peak (e.g., Node 7). This may attribute to the increased $SO_2$ emissions from household heating in winter (Liao et al., 2017). Like other primary pollutants, the local $SO_2$ emissions become trapped close to the surface under the stable nocturnal condition, resulting in a much higher nighttime peak compared to the pre-noon peak.

[revised manuscript text omitted]

**3.5 Evaluation against heavy polluted episodes**
**3.3 Quantifying the contribution of ABL anomaly to typical-month PM$_{2.5}$ air quality**

To improve air quality, the Chinese government promulgated "Air Pollution Prevention and Control Action Plan" in 2013. As a consequence, observed annual mean PM$_{2.5}$ concentrations decrease by about 37% over Beijing during 2013-2017. However, severe wintertime PM$_{2.5}$ pollution events still frequently wreaked havoc across Beijing and its surroundings, which resulted in severe damages to the environment and human health (Gao et al., 2017;Gao et al., 2015). It is therefore pressing to understand the factors affecting the occurrence of such serious PM$_{2.5}$ pollution. Previous studies highlighted potential importance of atmospheric conditions to the wintertime PM$_{2.5}$ air quality (Cai et al., 2017). Since the fraction of time for which the different atmospheric conditions dominate can vary from year to year, elucidation of the meteorological roles in those serious pollution periods has a significant importance. In this section, we evaluate the contribution of ABL anomaly to elevated PM$_{2.5}$ concentration in three typical pollution months, i.e., January of 2013, December of 2015 and December of 2016.

In 2013, the Chinese State Council released the "Atmospheric Pollution Prevention and Control Action Plan" to implement a megacity cluster-scale joint prevention and control strategy program. As a result, the PM$_{2.5}$ in Beijing decreased from 89.5 μg/m$^3$ in 2013 to 73.0 μg/m$^3$ in 2016. However, these meteorology-driven pollution episodes to some degree obscure the true impacts of the emission control strategies implemented by governmen t.

 Heavy PM$_{2.5}$  pollution episodes occurred frequently in January of 2013, December of 2015 and December of 2016  resulting in anomalously high month-averaged PM$_{2.5}$ concentrations in the Beijing urban area (180.1 μg/m$^3$, 151.8 μg/m$^3$ and 151.2 μg/m$^3$, respectively). Fig.7 shows the hourly PM$_{2.5}$ variationin the three selected months, along with daily ABL types.  In general, the pollution  episodes were  associated with  Nodes 3 and 9, and the clean  episodes corresponded to  Node 1. For example, the severe pollution episode that occurred from 9–14 January 2013 was due to the alternate  control of  Nodes 3 and 9, and the pollution episode from 15–21 December 2016 was related to the persistency of  Node 9. Conversely, multiday control of  Node 1 caused a clean episode from 14–16 December 2015. The linkages between PM$_{2.5}$ air quality and ABL type  are consistent with the previous long-term analyses , indicating that the changing ABL type is one of the primary drivers of day-to-day variations in wintertime PM$_{2.5}$ air quality over Beijing.

[Figure]

**Figure 7. Time series of hourly PM$_{2.5}$ concentrations in (a) January of 2013, (b) December of 2015, and (c) December of 2016. The daily ABL types (i.e., SOM nodes) are shown at the top of each plot (red numbers).**

~~The monthly PM$_{2.5}$ concentrations in the Beijing urban area reached up to 180.8 μg/m$^3$, 153.9 μg/m$^3$ and 147.9 μg/m$^3$ in January 2013, December 2015 and December 2016, respectively. All these values were far larger than the 4-yr winter mean PM$_{2.5}$ concentration (110.6 μg/m$^3$). Although the characteristics of PM$_{2.5}$ air quality depend on many complex elements, the major contributors are the pollutant emissions and meteorological conditions.In 2013, the Chinese State Council released the "Atmospheric Pollution Prevention and Control Action Plan" to implement a megacity cluster-scale joint prevention and control strategy program. As a result, the PM$_{2.5}$ in Beijing decreased from 89.5 μg/m$^3$ in 2013 to 73.0 μg/m$^3$ in 2016. However, these meteorology-driven pollution episodes to some degree obscure the true impacts of the emission control strategies implemented by government..8showsto the winter mean frequency (2013-2016) forthreepolluted4-yrmeanthe greatestoccurrences of nodes3 and 9 (the two most polluted types)the node1 (the clean~~

decreased during the three polluted months. For example, Node 9 occurrence was nearly trebled in

December of 2016, and total occurrence of Nodes 3 and 9 doubled in January of 2013. These results highlighted potential contribution of ABL anomaly to the elevated PM$_{2.5}$ concentrations in these polluted months.

[Figure]

**Figure 8. Occurrence frequency of the ABL types (i.e., SOM nodes) during (a) January 2013, (b) December 2015, and (c) December 2016. The winter-averaged frequency during the 5-year (2013–2017) period is repeated on each plot for comparison.**

Assuming the linkages between ABL type and PM$_{2.5}$ loading are constant in different years, the contribution of the anomalous ABL meteorology to PM$_{2.5}$ air quality can be estimated through a meteorology-to-environment method, which is revised from the circulation-to-environment method proposed by Zhang et al. (2012).

For each selected month

, we define the deviation in PM$_{2.5}$ from the 5-year winter-averaged concentration ($C_{WIN}$) as the total anomaly (C') is due to the combined effects of emission and meteorology. The anomaly calculated from the mean PM$_{2.5}$ loadings for nine ABL types and their occurrence frequencies during each month can be considered to represents the PM$_{2.5}$ change caused by the anomalous ABL meteorology.  The ABL-driven anomaly ($C_{ABL}'$) is calculated through $-\sum_i F_i \cdot C_i - C_{WIN}$, where $F_i$ is the occurrence frequency of type-$i$ ABL during a specific month and $C_i$ is the corresponding PM$_{2.5}$ loading featuring that type. The ratio of $C_{ABL}'$ to $C'$  is then used to assess the relative  contribution of the ABL anomaly to the total anomaly. The calculated results (Table 1) shows that the ABL-driven PM$_{2.5}$ changes are 44.4 µg/m$^3$ in January 2013, 22.2 µg/m$^3$ in December 2015, and 34.6 µg/m$^3$ in December 2016, which explain 58.3%, 46.4% and 73.3% of total anomaly in respective month.  These quantitative estimations demonstrate that  the elevated PM$_{2.5}$ concentrations during the three polluted months can be largely attributed to anomalous ABL conditions.

**Table 1. Estimated contribution of ABL anomaly to elevated PM$_{2.5}$ concentration in January of 2013, December of 2015 and December of 2016.**

| Pollution month | Month-averaged PM$_{2.5}$ concentration (µg/m$^3$) | Total anomaly $C'$ (µg/m$^3$) | ABL-driven anomaly $C_{ABL}'$ (µg/m$^3$) | Contribution ratio of ABL-driven anomaly (%) |
|---|---|---|---|---|
| January 2013 | 180.1 | 76.1 | 44.4 | 58.3 |
| December 2015 | 151.8 | 47.8 | 22.2 | 46.4 |
| December 2016 | 151.2 | 47.2 | 34.6 | 73.3 |

**4. Summary**

The influence of ABL meteorology on Beijing's air quality is relatively unclear due to the lack of long-term observations. Meanwhile, the long years of routine radiosondes remain underutilized as a tool for urban pollution studies. In this study, the SOM was applied to -yr (2013-2017)

radiosonde-based $\theta_v$ profile to classify the state of ABL  over Beijing. The  classified ABL types were then evaluated in relation to near-surface air quality , with an attempt to understand the roles of the  changing ABL  structure in  air quality variation in Beijing. The main findings are as follows:

1) The SOM provides a continuum of nine ABL types (i.e., SOM nodes), and each  is characterized with

   distinct  dynamic and thermodynamic conditions. within

   the ABL. Node 1 represent a near neutral layer with the lowest $\theta_v$ gradient and the highest wind speed. Node features a strong static stability in the upper ABL. In contrast, Node 9 and Node 7 respectively correspond

   to the moderate and strong static stability in the lower ABL.

2) The self-organized ABL types are capable of characterizing the influence of nocturnal mixing on near-surface

   pollutant loadings. From the near neutral (i.e.,  Node 1) to strong stable conditions (i.e.,

   Node 9), the  average concentrations of  $PM_{2.5}$, $NO_2$ and $CO$

   increased dramatically during all seasons except summer. Meanwhile,

   The diurnal  cycles of these pollutant species are strongly modulated  by

   ABL dynamics. Although the modulation effect varies from season to season, the higher peak

   concentrations commonly occur under the more stable conditions.

   However, increasing stability has

   opposite effect on $O_3$, resulting in the lowest $O_3$ level in Node 9. For $SO_2$, the highest average concentrations

   are typically observed in Node 3. The pre-noon $SO_2$ peaks are more significant after a strong stable night.

3)

4)

 Analysis of three typical wintertime  pollution months (i.e., January 2013, December 2015 and December 2016) suggests that the ABL types are one of the primary drivers of day-to-day PM$_{2.5}$ variations in Beijing. During the three pollution months, the  frequency of the nstable ABL types (i.e., Nodes 9 and 3) increases significantly compared  to the 5-ye (2013-2017) winter mean

3)Using a meteorology-to-environment method, the relative (absolute) contribution of the ABL anomaly to  elevated PM$_{2.5}$  concentrations  are estimated to be 58.3 % (44.4 μg/m$^3$)  in January 2013, 46.4 % (22.2 μg/m$^3$)  in December 2015, and 73.3 % (34.6 μg/m$^3$)  in December 2016.

This work revealed the common pattern of the ABL influence  on Beijing' air quality. The established  linkages between ABL type and air quality could be useful for developing an operational forecast and warning system. In addition, this work demonstrated that the SOM-based ABL classification scheme is a  helpful tool for understanding urban air pollution. Since the SOM technique is good at feature extraction, the coarse-resolution radiosonde s can be taken as  input to classify the state of the ABL. Therefore, the SOM-based ABL classification scheme can take advantage of the long-term available radiosondes, making it a simple and economical alternative  other approaches to stability classification. We believe that the pollution-related ABL research and the formulation of pollution control measures could benefit from application of the SOM analytical tool.

**Data availability**

The datasets used in this study are publicly available at the University of Wyoming (http://weather.uwyo.edu/), the Ministry of Environmental Protection of the People's Republic of China (http://datacenter.mep.gov.cn/), the U.S. Department of State Air Quality Monitoring Program (http://www.stateair.net/).

**Competing interests**

The authors declare no conflict of interest.

**Acknowledgements**

This study is supported by the National Key Research and Development Plan of China (Nos. 2017YFC0209606 and 2016YFC0203305), the National Natural Science Foundation of China (Nos. 41630422, 41475140 and 41475004) and the Special Fund for Basic Scientific Research Business of Central Public Research Institutes (PM-zx703-201601-019). The authors would like to thank the Beijing Meteorological Bureau, the Ministry of Environmental Protection of the People's Republic of China and the Wyoming Weather Web for providing related data.

[revised manuscript text omitted]

**T, F, and C represent the total, fine-, and coarse-mode particles.**